# PROTECT THE WEAK: CLASS-FOCUSED ONLINE LEARNING FOR ADVERSARIAL TRAINING

## ABSTRACT

Adversarial training promises a defense against adversarial perturbations in terms of average accuracy. In this work, we identify that the focus on the average accuracy metric can create vulnerabilities to the "weakest" class. For instance, on CIFAR10, where the average accuracy is 47%, the worst class accuracy can be as low as 14%. The performance sacrifice of the weakest class can be detrimental for real-world systems, if indeed the threat model can adversarially choose the class to attack. To this end, we propose to explicitly minimize the worst class error, which results in a min-max-max optimization formulation. We provide high probability convergence guarantees of the worst class loss for our method, dubbed as *class focused online learning* (CFOL), using techniques from the online learning community. CFOL can be plugged into existing training setups with virtually no overhead in computation. We observe significant improvements on the worst class accuracy of 30% for CIFAR10. We also observe consistent behavior across CIFAR100 and STL10. Intriguingly, we find that minimizing the worst case can even sometimes improve the average.

## 1 INTRODUCTION

The susceptibility of neural networks to adversarial noise (Goodfellow et al., 2014; Szegedy et al., 2013) has been a grave concern over the launch of such systems in real-world applications. Several techniques defending such attacks that optimize the average performance have been proposed (Papernot et al., 2016; Raghunathan et al., 2018; Guo et al., 2017; Madry et al., 2017; Zhang et al., 2019). In response, even stronger attacks have been proposed (Carlini & Wagner, 2016; Engstrom et al., 2018; Carlini, 2019). Indeed, recent studies demonstrate that regardless of the defense there exists an attack that can lower the average performance of the system (Shafahi et al., 2018).

In this work, we argue that the average performance is not the only criterion that is of interest for real-world applications. For classification, in particular, optimizing the average performance provides no guarantees for the "weakest" class. This is critical in scenarios where an attacker can pick the class adversarially in addition to the adversarial perturbation. It turns out that the worst performing class can indeed be much worse than the average in adversarial training. This difference is already present in clean training but we critically observe, that the gap between the average and the worst, is greatly exacerbated in adversarial training. This gap can already be observed on CIFAR10 where the accuracy across classes is far from uniform with 47% average robust accuracy while the worst class is 14% (see Figure 1). The effect is even more prevalent when more classes are present as in CIFAR100 where we observe that the worst class has *zero* accuracy while the best has 70% (see Appendix C where we include multiple other datasets). Despite the focus on adverarial training, we note that the same effect can be observed for robust evaluation after *clean* training (see Figure 4).

This dramatic drop in accuracy for the weakest classes begs for different approaches than the classical empirical risk minimization (ERM), which focuses squarely on the average loss. We suggest a simple alternative, which we dub *class focused online learning* (CFOL), that can be plugged into existing adversarial training procedures. Instead of minimizing the average performance over the dataset we sample from an adversarial distribution over classes that is learned jointly with the model parameters. In this way we aim at becoming robust to an attacker that can adversarially *choose the class*. The focus of this paper is thus on the robust accuracy of the weakest classes instead of the average robust accuracy.

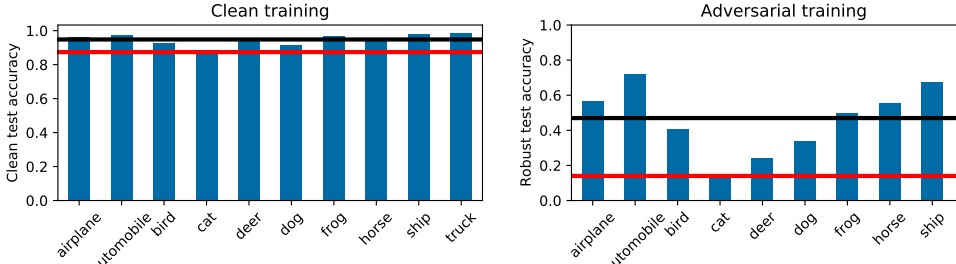

Figure 1: The error across classes is already not perfectly uniform in clean training on CIFAR10. However, this phenomenon is significantly worsened in adversarial training when considering the robust accuracy. That is, some classes perform much worse than the average. The worst class accuracy and average accuracy is depicted with a red and black line respectively.

Concretely, we make the following contributions:

- We propose a simple solution which relies on the classical bandit algorithm from the online learning literature, namely the Exponential-weight algorithm for Exploration and Exploitation (Exp3) (Auer et al., 2002). The method is directly compatible with standard adversarial training procedures (Madry et al., 2017), by replacing the empirical distribution with an adaptively learned adversarial distribution over classes.
- We carry out extensive experiments comparing CFOL against three strong baselines across three datasets, where we consistently observe that CFOL improves the weakest classes.
- We support the empirical results with high probability convergence guarantees for the worst class accuracy and establish direct connection with the conditional value at risk (CVaR) (Rockafellar et al., 2000) uncertainty set from the distributional robust optimization.

**Overview**  We define our new objective in Section 2, followed by the description of the proposed method in Section 3. In Section 3.1 we prove convergence guarantees. We then move to the empirical work in Section 4, where we observe and resolve the issue of robust overfitting in the context of the worst class. Our main empirical findings are covered in Section 5 and we conclude in Section 6.

## 1.1 RELATED WORK

**Adversarial examples**  Goodfellow et al. (2014); Szegedy et al. (2013) are the first to make the important observation that deep neural networks are vulnerable to small adversarially perturbation of the input. Since then, there has been a growing body of literature addressing this safety critical issue, spanning from certified robust model (Raghunathan et al., 2018), distillation (Papernot et al., 2016), input augmentation (Guo et al., 2017), to adversarial training (Madry et al., 2017; Zhang et al., 2019). We focus on adversarial training in this paper. While certified robustness is desirable, adversarial training remains one of the most successful defenses in practice.

In a concurrent work Tian et al. (2021) independently observe the non-uniform accuracy over classes in adversarial training, further strengthening the case that lack of class-wise robustness is indeed an issue. However, they mainly focus on constructing an attack that can enlarge this disparity.

**Minimizing the maximum**  Connected to our method is that of focused online learning (FOL) (Shalev-Shwartz & Wexler, 2016), which similarly takes a bandit approach, but instead re-weights the distribution over the $N$ training examples, independent of the class label. This naturally leads to a convergence rate in terms of the number of examples $N$ instead of the number of classes $k$ for which usually $k \ll N$. We compare in more detail theoretically and empirically in Section 3.1 and Section 5 respectively.

Interpolations between average and maximum loss have been considered in various other settings: for class imbalanced datasets (Lin et al., 2017), in federated learning (Li et al., 2019), and more generally the tilted empirical risk minimization (Li et al., 2020; Lee et al., 2020).

**Distributional robust optimization**   The accuracy over the worst class can be seen as a particular re-weighing the data distribution which adversarially assigns all weight to a single class. Worst case perturbation of the data distribution have more generally been studied under the framework of distributional robust stochastic optimization (DRO) (Ben-Tal et al., 2013; Shapiro, 2017). Instead of attempting to minimizing the empirical risk on a training distribution $P_0$, this framework considers some *uncertainty set* around the training distribution $\mathcal{U}(P_0)$ and seeks to minimize the worst case risk within this set, $\sup_{Q \in \mathcal{U}(P_0)} \mathbb{E}_{x \sim Q}[\ell(x)]$.

A choice of uncertainty set, which has been given significant attention in the community, is conditional value at risk (CVaR), which aims at minimizing the weighted average of the tail risk (Rockafellar et al., 2000; Levy et al., 2020; Kawaguchi & Lu, 2020; Fan et al., 2017; Curi et al., 2019). CVaR has been specialized to a re-weighting over class labels, namely labeled conditional value at risk (LCVaR) (Xu et al., 2020). This was originally derived in the context of imbalanced dataset to re-balance the classes. It is still applicable in our setting and we thus provide a comparison. The original empirical work of (Xu et al., 2020) only considers the full-batch setting. We complement this by demonstrating LCVaR in a stochastic setting.

In Duchi et al. (2019); Duchi & Namkoong (2018) they are, similarly to our setting, interested in uniform performance over various groups. However, these groups are assumed to be *latent* subpopulations, which introduces significant complications. It is thus concerned with a different setting, an example being training on a dataset implicitly consisting of multiple text corpora.

CFOL can also be formulated in the framework of DRO by choosing an uncertainty set that can re-weight the $k$ class-conditional risks. The precise definition is given in Appendix B.1. We further establish a direct connection between the uncertainty sets of CFOL and CVaR that we make precise in Appendix B.1, which also contains a summary of the most relevant related methods in Table 4.

## 2   PROBLEM FORMULATION AND PRELIMINARIES

**Notation**   The underlying data distribution is denoted by $\mathcal{D}$ with examples $x \in \mathbb{R}^d$ and classes $y \in [k]$. A given iteration is characterized by $t \in [T]$, while $p_t^y$ indicates the $y^{\text{th}}$ index of the $t^{\text{th}}$ iterate. The indicator function is denoted with $\mathbb{1}_{\{\text{boolean}\}}$ and $\text{unif}(n)$ indicates the uniform distribution over $n$ elements. An overview of the notation is provided in Appendix A.

In classification, we are normally interested in minimizing the population risk $\mathbb{E}_{(x,y) \sim \mathcal{D}}[\ell(\theta, x, y)]$ over our model parameters $\theta \in \mathbb{R}^p$, where $\ell$ is some loss function of $\theta$ and example $x \in \mathbb{R}^d$ with an associated class $y \in [k]$. Madry et al. (2017) formalized adversarially training by modifying this objective with an adversarial perturbation to each example. That is, we instead want to find a parameterization $\theta$ of our predictive model which solves the following optimization problem:

$$\min_{\theta} L(\theta) := \mathbb{E}_{(x,y) \sim \mathcal{D}} \left[ \max_{\delta \in \mathcal{S}} \ell(\theta, x + \delta, y) \right], \tag{1}$$

where each $x$ is now perturbed by adversarial noise $\delta \in \mathcal{S} \subseteq \mathbb{R}^d$. Common choices of $\mathcal{S}$ include norm-ball constraints as in Madry et al. (2017) or bounding some notion of perceptual distance (Laidlaw et al., 2020). When the distribution over classes is uniform this is implicitly minimizing the *average* loss over all class. This does not guarantee high accuracy for the *worst* class as illustrated in Figure 1, since we only know with certainty that $\max \geq \text{avg}$.

Instead, we will focus on a different objective, namely minimizing the *worst class-conditioned risk*:

$$\min_{\theta} \max_{y \in [k]} \left\{ L_y(\theta) := \mathbb{E}_{x \sim p_{\mathcal{D}}(\cdot|y)} \left[ \max_{\delta \in \mathcal{S}} \ell(\theta, x + \delta, y) \right] \right\}. \tag{2}$$

This follows the philosophy that "*a chain is only as strong as its weakest link*". In our particular setting it models a scenario where the attacker can adversarially choose what class the model is evaluated on. In safety critical application, such as autonomous driving, modeling even just a single class wrong can still have catastrophic consequences.

As the maximum in Equation 2 is a discrete maximization problem its treatment requires more care. We will take a common approach and construct a convex relaxation to this problem in Section 3.

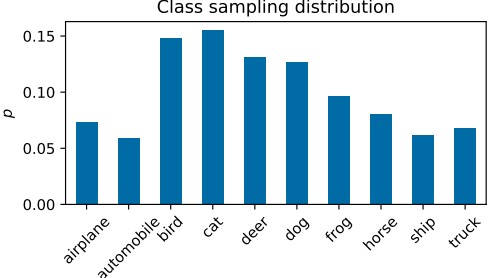

Figure 2: Contrary to ERM, which samples the examples uniformly, CFOL samples from an adaptive distribution. The learned adversarial distribution is non-uniform over the classes in CIFAR10 when using adversarial training. As expected, the hardest classes are also most frequently sampled.

## 3  METHOD

Since we do not have access to the true distribution $\mathcal{D}$, we will instead minimize over the provided empirical distribution. Let $\mathcal{N}_y$ be the set of data point indices for class $y$ such that the total number of examples is $N = \sum_{y=1}^{k} |\mathcal{N}_y|$. Then, we are interested in minimizing the *maximum empirical class-conditioned risk*,

$$\max_{y \in [k]} \widehat{L}_y(\theta) := \frac{1}{|\mathcal{N}_y|} \sum_{i \in \mathcal{N}_y} \max_{\delta \in \mathcal{S}} \ell(\theta, x_i + \delta, y). \tag{3}$$

We relax this discrete problem to a continuous problem over the simplex $\Delta_k$,

$$\max_{y \in [k]} \widehat{L}_y(\theta) \leq \max_{p \in \Delta_k} \sum_{y=1}^{k} p_y \widehat{L}_y(\theta). \tag{4}$$

Note that equality is attained when $p$ is a dirac on the argmax over classes.

Equation 4 leaves us with a two-player zero-sum game between the model parameters $\theta$ and the class distribution $p$. From the perspective of $p$ it is just a linear objective, albeit adversarially picked by the model. This immediately let us use a no-regret algorithm under simplex constraint, namely Hedge (Freund & Schapire, 1997):

$$w_y^t = w_y^{t-1} - \eta \widehat{L}_y(\theta^t), \quad p_y^t = \exp\left(w_y^t\right) / \sum_{y=1}^{k} \exp\left(w_y^t\right). \tag{Hedge}$$

To show convergence for (Hedge) the loss needs to satisfy certain assumptions. In our case of classification, the loss is the zero-one loss $\ell(\theta, x, y) = \mathbb{1}[h_\theta(x) \neq y]$, where $h_\theta(\cdot)$ is the predictive model. Hence, the loss is bounded, which is a sufficient requirement.

Note that (Hedge) relies on zero-order information of the loss, which we indeed have available. However, in the current form, (Hedge) requires so called *full information* over the $k$ dimensional loss vector. In other words, we need to compute $\widehat{L}_y(\theta)$ for all $y \in [k]$ at every iteration. When accessing the data through mini-batches, it is strictly not possible to compute even an estimate for each $y$ if the number of classes is larger than the batch size.

Following the seminal work of Auer et al. (2002), we instead construct an unbiased estimator of the $k$ dimensional loss vector $\widehat{L}(\theta^t) := (\widehat{L}_1(\theta^t), ..., \widehat{L}_k(\theta^t))^\top$ based on a sampled class $y^t$ from some distribution $y^t \sim p^t$. This stochastic formulation further lets us estimate the class conditioned risk $\widehat{L}_{y^t}(\theta^t)$ with an unbiased sample $i \sim \text{unif}(|\mathcal{N}_{y^t}|)$. This leaves us with the following estimator,

$$\widetilde{L}_y^t = \begin{cases} L_{y,i}(\theta^t)/p_y^t & y = y^t \\ 0 & \text{otherwise} \end{cases} \tag{5}$$

---

**Algorithm 1:** Class focused online learning (CFOL)

---

Algorithm parameters: a step rule $\mathrm{ModelUpdate}$ for the model satisfying Assumption 1,
  adversarial step-size $\eta > 0$, uniform mixing parameter $\gamma = 1/2$, and the loss $L_{y,i}(\theta)$.

Initialization: Set $w^0 = 0$ such that $q^0$ and $p^0$ are uniform.

**foreach** $t$ *in* $0..T$ **do**

$\quad\begin{aligned}
& y^t \sim p^t \; ; && \text{// sample class} \\
& i^t \sim \mathrm{unif}(|\mathcal{N}_{y^t}|) \; ; && \text{// sample uniformly from class} \\
& \theta^{t+1} = \mathrm{ModelUpdate}(\theta^t, L_{y^t,i^t}(\theta^t)) \; ; && \text{// update model parameters} \\
& \widetilde{L}_y^t = \mathbb{1}_{\{y=y^t\}} L_{y,i^t}(\theta^t)/p_y^t \; \forall y \; ; && \text{// construct estimator} \\
& w_y^{t+1} = w_y^t - \eta \widetilde{L}_y^t \; \forall y \; ; && \text{// update the adv. class distribution} \\
& q_y^{t+1} = \exp\left(w_y^{t+1}\right) / \sum_{y=1}^k \exp\left(w_y^{t+1}\right) \; \forall y; \\
& p_y^{t+1} = \gamma \frac{1}{k} + (1-\gamma) q_y^{t+1} \; \forall y;
\end{aligned}$

**end foreach**

---

where $L_{y,i}(\theta) := \max_{\delta \in \mathcal{S}} \ell(\theta, x_i + \delta, y)$. It is easy to verify that this estimator is unbiased,

$$\mathbb{E}_{y \sim p}\left[\widetilde{L}_y^t\right] = \left(1 - p_y^t\right) \cdot 0 + p_y^t \cdot \frac{\widehat{L}_y(\theta^t)}{p_y^t} = \widehat{L}_y(\theta^t). \tag{6}$$

For ease of presentation the estimator only uses a single sample but this can trivially be extended to a mini-batch where classes are drawn i.i.d. from $p^t$.

We could pick $p^t$ to be the learned adversarial distribution, but it is well known that this can lead to unbounded regret if some $p_y^t$ is small (see Appendix A for definition of regret). Auer et al. (2002) resolves this problem with Exp3, which instead learns a distribution $q^t$, then mixes it with a uniform distribution, to eventually sample from $p_y^t = \gamma \frac{1}{k} + (1 - \gamma) q_y^t$ where $\gamma \in (0, 1)$. Intuitively, this enforces exploration. In the general case this $\gamma$ needs to be picked carefully and small enough, but similarly to Shalev-Shwartz & Wexler (2016), we show that $\gamma = 1/2$ suffice in our setting.

Our algorithm thus updates $q^{t+1}$ with (Hedge) using the estimator $\widetilde{L}^t$ with sample drawn from $p^t$ and subsequently plays $p^{t+1}$. CFOL in conjunction with the simultaneous update of the minimization player can be found in algorithm 1 with an example of a learned distribution $p^t$ in Figure 2.

Practically, the scheme bears negligible computational overhead over ERM since the softmax required to sample is of the same dimensionality as the softmax used in the forward pass through the model. This computation is negligible in comparison with backpropagating through the entire model. For further details on the implementation we refer to Appendix C.4

### 3.1 CONVERGENCE RATE

To understand what kind of result we can expect, it is worth entertaining a hypothetical worst case scenario. Imagine a classification problem where one class is much harder to model than the remaining classes. We would expect the learning algorithm to require exposure to examples from the hard class in order to model that class appropriately—otherwise the classes would not be distinct. From this one can see why ERM might be slow. The algorithm would naively pass over the entire dataset in order to improve the hard class using only the fraction of examples belonging that class. In contrast, if we can adaptively focus on the difficult class, we can avoid spending time on classes that are already improved sufficiently. As long as we can adapt fast enough, as expressed through the regret of the adversary, we should be able to improve on the convergence rate for the worst class.

We will now make this intuition precise by establishing a high probability convergence guarantee for the worst class loss analogue to that of FOL. For this we will assume that the model parameterized by $\theta$ enjoys a so called mistake bound of $C$ (Shalev-Shwartz et al., 2011, p. 288). The proof is deferred to appendix A.

**Assumption 1.** For any sequence of classes $(y^1, ..., y^T) \in [k]^T$ and class conditioned indices $(i^1, ..., i^T)$ with $i^t \in \mathcal{N}_{y^t}$ the model enjoys the following bound for some $C' < \infty$ and $C = \max\{k \log k, C'\}$,

$$\sum_{t=1}^{T} L_{y^t, i^t}(\theta^t) \leq C. \tag{7}$$

*Remark* 1. The requirement $C \geq k \log k$ will be needed to satisfy the mild step-size requirement $\eta \leq 2k$ in Lemma 1. In most settings the smallest $C'$ is some fraction of the number of iterations $T$, which in turn is much larger than the number of classes $k$, so $C = C'$.

With this at hand we are ready to state the convergence of the worst class-conditioned empirical risk.

**Theorem 1.** *If algorithm 1 is run on bounded rewards $L_{y^t, i^t}(\theta^t) \in [0, 1]$ $\forall t$ with step-size $\eta = \sqrt{\log k/(4kC)}$, mixing parameter $\gamma = 1/2$ and the model satisfies Assumption 1, then after $T$ iterations with probability at least $1 - \delta$,*

$$\max_{y \in [k]} \frac{1}{n} \sum_{j=1}^{n} \widehat{L}_y(\theta^{t_j}) \leq \frac{6C}{T} + \frac{\sqrt{4k \log(2k/\delta)}}{\sqrt{T}} + \frac{(1 + 2k) \log(2k/\delta)}{3T} + \frac{\sqrt{2 \log(2k/\delta)}}{\sqrt{n}} + \frac{2 \log(2k/\delta)}{3n}, \tag{8}$$

*for an ensemble of size $n$ where $t_j \stackrel{iid}{\sim} \mathrm{unif}(T)$ for $j \in [n]$.*

To contextualize Theorem 1, let us consider the simple case of linear binary classification mentioned in Shalev-Shwartz & Wexler (2016). In this setting SGD needs $\mathcal{O}(CN)$ iterations to obtain a consistent hypothesis. In contrast, the iteration requirement of FOL decomposes into a sum $\widetilde{\mathcal{O}}(C + N)$ which is much smaller since both $C$ and $N$ are large. When we are only concerned with the convergence of the worst class we show that CFOL can converge as $\widetilde{\mathcal{O}}(C + k)$ where usually $k \ll C$. Connecting this back to our motivational example in the beginning of this section, this sum decomposition exactly captures our intuition. That is, adaptively focusing the class distribution can avoid the learning algorithm from needlessly going over all $k$ classes in order to improve just one of them.

From Theorem 1 we also see that we need an ensemble of size $n = \Omega(\sqrt{\log(k/\delta)}/\varepsilon^2)$, which has only mild dependency on the number of classes $k$. If we wanted to drive the worst class error $\varepsilon$ to zero the dependency on $\varepsilon$ would be problematic. However, for adversarial training, even in the best case of the CIFAR10 dataset, the *average* error is larger than $1/2$. We can expect the worst class error to be even worse, so that only a small $n$ is required. In practice, a single model turns out to suffice.

Note that the maximum upper bounds the average, so by minimizing this upper bound as in Theorem 1, we are implicitly still minimizing the usual average loss. In addition, Theorem 1 shows that a mixing parameter of $\gamma = 1/2$ is sufficient for minimizing the worst class loss. Effectively, the average loss is still directly being minimized, but only through half of the sampled examples.

## 4 EXPERIMENTAL SETUP: OVERCOMING ROBUST OVERFITTING

Robust overfitting is a well-documented challenge in adversarial training (Rice et al., 2020; Pang et al., 2020). This phenomenon is characterized by a rapid drop in the validation accuracy shortly after the first piecewise step-size decay. The reported models in the literature are indeed early stopped prior to the second step-size decay (e.g., in Madry et al. (2017); Zhang et al. (2019)). This is particularly problematic when interested in the worst class accuracy, which might not be aligned with the average performance across training epochs. Below, we conduct preliminary experiments on CIFAR10 confirming this problem and propose a solution. We adopt the setup for adversarial training with PGD in Madry et al. (2017) with the stepsize scheduling from Zhang et al. (2019). We consider adversarial training on the empirical distribution (ERM-AT) and adversarial training on the jointly learned adversarially class distribution (CFOL-AT).

With ERM-AT the worst class has only $14\%$ robust test accuracy when early stopping based on the average accuracy. Early stopping based on the worst class can improve on the worst class accuracy, but only marginally, as we show in the supplementary (see Table 5). This minor improvement comes at a huge cost for the average accuracy which drops from $47\%$ to $38\%$. Ideally both metrics should instead increase monotonically.

Table 1: Robust accuracy on CIFAR10 under early stopping without temporal ensembling. The number in parenthesis indicates the epoch of piecewise constant step-size decay. For CFOL-AT to consistently converge it was necessary to decay earlier than the usual decay at epoch 75. Fortunately, known regularization techniques mitigate this problem which leads to our main results in Section 5.

|  | **ERM-AT (75)** | **ERM-AT (33)** | **CFOL-AT (33)** |
|---|---|---|---|
| Average | **0.47** | 0.45 | 0.42 |
| Worst class | 0.14 | 0.13 | **0.27** |

For CFOL-AT the problem of robust overfitting presents itself differently. The worst class accuracy improves much more rapidly, reaching $27\%$ even when early stopped based on the average accuracy. Note that this is not surprising and is in line with the convergence rate derived in Section 3.1. The implication is that we are required to run CFOL-AT for a shorter time to get consistent performance and carefully select the step-size decay point. This limited training time, due to the earlier overfitting, unfortunately restricts the expressivity of the model. We show comparison with both ERM-AT with similar early step-size decay and also more standard step-size decay of Zhang et al. (2019) in Table 1. Although we see a significant improvement from $14\%$ to $27\%$ in terms of the worst class we ideally want the method be stable for more iterations.

To avoid the root problem of overfitting, we regularize the training objective. We adopt temporal ensembling (TE) which is a regularization technique that was shown to mitigate overfitting in adversarial training (Laine & Aila, 2016; Dong et al., 2021). This allows us to reliably run CFOL-AT under exactly the same training configuration as ERM-AT and thus prevents further hyperparameter tuning. Additionally, TE leads to a more fair comparison with ERM-AT, which enjoys an improvement in terms of the worst class robust accuracy from $14\%$ to $21\%$. We make use of this additional regularization in all subsequent experiments. In this work we focus on the robust accuracy. However, we add a cautionary note, since we observe that TE can lead to a reduction in clean accuracy (see Table 6).

## 5 EXPERIMENTS

We consider the common adversarial setting where the constraint set of the attacker $\mathcal{S}$ is $\ell_\infty$-bounded attacks. We test on three datasets with different dimensionality, number of examples per class and number of classes. Specifically, we consider CIFAR10, CIFAR100 and STL10 (Krizhevsky et al., 2009; Coates et al., 2011) (see Appendix C.2 for further details).

**Hyper-parameters** Unless otherwise noted, we use the standard adversarial training setup of a ResNet50 network (He et al., 2016) with a learning rate $\tau = 0.1$, momentum of $0.9$, weight decay of $5 \cdot 10^{-4}$, batch size of $128$ (Madry et al., 2017) with a piece-wise constant weight decay of $0.1$ at epoch 75 and 90 for a total of 100 epochs according to Zhang et al. (2019), as well as regularizing with temporal ensembling as explained in Section 4. For the attack we similarly adopt the common attack radius of $8/255$ using 7 steps of projected gradient descent (PGD) with a step-size of $2/255$ (Madry et al., 2017). For STL10 we adopt the learning rate $\tau = 0.01$ and use a batch-size of $64$ to fit it inside our hardware. For evaluation we use the stronger attack of 20 step PGD.

**Baselines** With this setup we compare our proposed method, CFOL, against empirical risk minimization (ERM), labeled conditional value at risk (LCVaR) (Xu et al., 2020) and focused online learning (FOL) (Shalev-Shwartz & Wexler, 2016). We add the suffix "AT" to all methods to indicate that the training examples are adversarially perturbed according to adversarial training of Madry et al. (2017). We consider ERM-AT as the core baseline, while we also implement FOL-AT and LCVaR-AT as alternative methods that can improve the worst performing class. For fair comparison, and to match existing literature, we do early stopping based on the average robust accuracy on the hold-out set. More details on hyperparameters and implementation can be found in Appendix C.1 and Appendix C.4 respectively. In Table 10 we additionally provide experiments for a variant of CFOL-AT which instead reweighs the gradients.

**Metrics** We report the average accuracy, the worst class accuracy and the accuracy across the 20% worst classes (referred to as the 20% tail) for both clean ($\mathrm{acc}_{\mathrm{clean}}$) and robust accuracy ($\mathrm{acc}_{\mathrm{rob}}$). The

mean and standard deviation in all tables are computed over 5 independent executions. We note that the aim is not to be state-of-the-art but rather provide a fair comparison between the methods.

The first core experiment is conducted on CIFAR10. In Table 2 the quantitative results are reported with the accuracy per class illustrated in Figure 3. The results reveal that all methods other than ERM-AT improve the worst performing class with CFOL-AT obtaining higher accuracy in the weakest class than all methods in both the clean and the robust case. Interestingly, FOL-AT performs better than LCVaR-AT, so for the next experiments we simply consider FOL as the non-ERM comparison method.

The results for the remaining datasets can be found in Table 3. The results exhibit similar patterns to the experiment on CIFAR10, where CFOL-AT improves both the worst performing class and the 20% tail. In the supplementary, we also provide results for early stopped models using the best worst class accuracy from the validation set (see Table 7), larger attack radius (see Table 8), and a different test time attack (see Table 9). In all cases, we observe that CFOL-AT has *consistently* improved accuracy with respect to the weakest class (and the 20% tail). We note that early stopping based on the worst class is not sufficient to make ERM-AT competitive with CFOL-AT.

We find that CFOL-AT consistently improves the accuracy for both the worst class and the 20% tail across the three datasets. At the same time, the average performance only suffers a minor reduction. The improvement seem to be even more noticeable in terms of the clean accuracy. Interestingly, FOL-AT improves even the *average* accuracy on CIFAR10 while having small variance, suggestion that sometimes focusing on the worst cases can improve the average.

Table 2: Accuracy on CIFAR10. For both clean test accuracy ($\mathrm{acc_{clean}}$) and robust test accuracy ($\mathrm{acc_{rob}}$) we report the average, 20% worst classes and the worst class. We compare our method (CFOL-AT) with standard adversarial training (ERM-AT) and two baselines (LCVaR-AT and FOL-AT). CFOL-AT significantly improves the robust accuracy for both the worst class and the 20% tail, while only incurring a small reduction in the average robust accuracy in comparison with ERM-AT.

|  |  | ERM-AT | CFOL-AT | LCVaR-AT | FOL-AT |
|---|---|---|---|---|---|
| $\mathrm{acc_{clean}}$ | Average | $0.74 \pm 0.01$ | $0.75 \pm 0.00$ | $0.75 \pm 0.01$ | $\mathbf{0.79} \pm 0.01$ |
|  | 20% tail | $0.52 \pm 0.03$ | $\mathbf{0.66} \pm 0.01$ | $0.54 \pm 0.02$ | $0.61 \pm 0.03$ |
|  | Worst class | $0.48 \pm 0.04$ | $\mathbf{0.63} \pm 0.02$ | $0.51 \pm 0.03$ | $0.56 \pm 0.03$ |
| $\mathrm{acc_{rob}}$ | Average | $0.47 \pm 0.02$ | $0.46 \pm 0.00$ | $0.46 \pm 0.03$ | $\mathbf{0.50} \pm 0.00$ |
|  | 20% tail | $0.24 \pm 0.02$ | $\mathbf{0.31} \pm 0.02$ | $0.23 \pm 0.02$ | $0.28 \pm 0.02$ |
|  | Worst class | $0.21 \pm 0.02$ | $\mathbf{0.30} \pm 0.01$ | $0.20 \pm 0.01$ | $0.22 \pm 0.03$ |

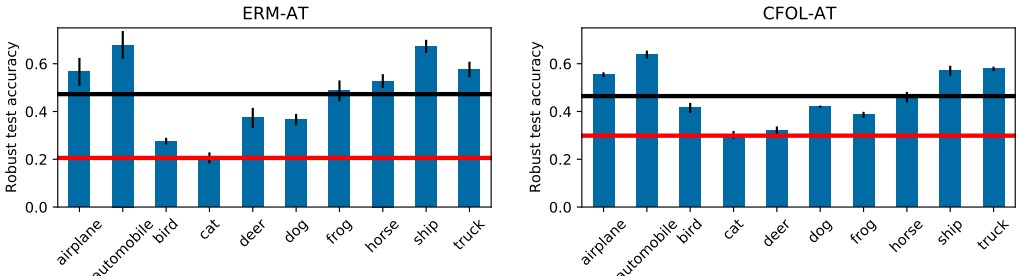

Figure 3: The robust test accuracy for CFOL-AT and ERM-AT over classes. The horizontal black and red line depicts the average and worst class accuracy over the classes respectively. The improvement in the minimum accuracy is notable when using CFOL-AT, while there is only marginal difference in the average accuracy.

Table 3: Clean test accuracy ($\text{acc}_{\text{clean}}$) and robust test accuracy ($\text{acc}_{\text{rob}}$) on CIFAR100 and STL10. We compare our method (CFOL-AT) with standard adversarial training (ERM-AT) and two baselines (LCVaR-AT and FOL-AT). CFOL-AT consistently improves the worst class accuracy as well as the $20\%$ worst tail.

|  |  |  | ERM-AT | CFOL-AT | FOL-AT |
|---|---|---|---|---|---|
| CIFAR100 | $\text{acc}_{\text{clean}}$ | Average | **0.54** $\pm$ 0.02 | 0.52 $\pm$ 0.05 | 0.53 $\pm$ 0.01 |
|  |  | 20% tail | 0.30 $\pm$ 0.04 | **0.35** $\pm$ 0.06 | 0.31 $\pm$ 0.04 |
|  |  | Worst class | 0.12 $\pm$ 0.04 | **0.19** $\pm$ 0.04 | 0.17 $\pm$ 0.04 |
|  | $\text{acc}_{\text{rob}}$ | Average | **0.27** $\pm$ 0.00 | 0.24 $\pm$ 0.02 | 0.26 $\pm$ 0.00 |
|  |  | 20% tail | 0.07 $\pm$ 0.02 | **0.09** $\pm$ 0.03 | 0.07 $\pm$ 0.02 |
|  |  | Worst class | 0.01 $\pm$ 0.01 | **0.03** $\pm$ 0.02 | 0.01 $\pm$ 0.01 |
| STL10 | $\text{acc}_{\text{clean}}$ | Average | 0.55 $\pm$ 0.03 | **0.55** $\pm$ 0.02 | 0.54 $\pm$ 0.02 |
|  |  | 20% tail | 0.25 $\pm$ 0.10 | **0.42** $\pm$ 0.08 | 0.27 $\pm$ 0.07 |
|  |  | Worst class | 0.23 $\pm$ 0.09 | **0.38** $\pm$ 0.07 | 0.22 $\pm$ 0.02 |
|  | $\text{acc}_{\text{rob}}$ | Average | **0.35** $\pm$ 0.01 | 0.34 $\pm$ 0.01 | 0.35 $\pm$ 0.01 |
|  |  | 20% tail | 0.09 $\pm$ 0.03 | **0.18** $\pm$ 0.03 | 0.11 $\pm$ 0.04 |
|  |  | Worst class | 0.07 $\pm$ 0.03 | **0.16** $\pm$ 0.03 | 0.07 $\pm$ 0.01 |

## 6 CONCLUSION

In this work, we have introduced a method for class focused online learning (CFOL), which samples from an adversarial learned distribution over classes. We establish high probability convergence results of the worst class for CFOL through a specialized regret analysis. In the context of adversarial examples this is motivated by an adversarial threat model in which the attacker chooses what class to evaluate on in addition to the perturbation. We conduct a thorough empirical validation on three datasets. The empirical results for adversarial training consistently demonstrate the improvement over the weakest classes. The work opens up for multiple interesting research avenues. Firstly, our method can also be applied in non-adversarial settings, and in the future we intend to look further into the challenging cases of non-uniform distribution over classes. Secondly, it remains open to establish generalization bounds for CFOL. Finally, it is interesting to understand *why* adversarial perturbations leads to inhomogeneous accuracies over classes.

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

## A CONVERGENCE ANALYSIS

### A.1 PRELIMINARY AND NOTATION

Consider the abstract online learning problem where for every $t \in [T]$ the player chooses an action $x^t$ and subsequently the environment reveals the loss function $l(\cdot, y^t)$. As traditional in the online learning literature we will measure performance in terms of regret, which compares our sequence of choices $\{x^t\}_{t=1}^T$ with a fixed strategy in hindsight $u$,

$$\mathcal{R}_T(u) = \sum_{t=1}^T l(x^t, y^t) - \min_x \sum_{t=1}^T l(u, y^t). \tag{9}$$

If we, instead of minimizing over losses $l$, maximize over rewards $r$ we define regret as,

$$\mathcal{R}_T(u) = \max_x \sum_{t=1}^T r(u, y^t) - \sum_{t=1}^T r(x^t, y^t). \tag{10}$$

When we allow randomized strategy, as in the bandit setting, $\mathcal{R}_T$ becomes a random variable that we wish to upper bound with high probability. For convenience we include an overview over the notation defined and used in Section 2 and Section 3 below.

| | |
|---|---|
| $k$ | Number of classes |
| $T$ | Number of iterations |
| $p_y^t$ | The $y^{\text{th}}$ index of the $t^{\text{th}}$ iterate |
| $\mathbb{1}_{\{\text{boolean}\}}$ | The indicator function |
| $\text{unif}(n)$ | Discrete uniform distribution over $n$ elements |
| $\mathcal{N}_y$ | Set of data point indices for class $y \in [k]$ |
| $N = \sum_{y=1}^k \lvert \mathcal{N}_y \rvert$ | The size of the data set |
| $L_{y,i}(\theta) := \max_{\delta \in \mathcal{S}} \ell(\theta, x_i + \delta, y)$ | Loss on a particular example |
| $\widehat{L}_y(\theta) := \frac{1}{\lvert \mathcal{N}_y \rvert} \sum_{i \in \mathcal{N}_y} L_{y,i}(\theta)$ | The empirical class-conditioned risk |
| $\widehat{L}(\theta) := (\widehat{L}_1(\theta), \ldots, \widehat{L}_k(\theta))^\top$ | The vector of all empirical class-conditioned risks |
| $L_y^t := L_{y,i}(\theta^t)$ | Class-conditioned estimator at iteration $t$ with $i \sim \text{unif}(\lvert \mathcal{N}_y \rvert)$ |

### A.2 CONVERGENCE RESULTS

We restate Algorithm 1 while leaving out the details of the classifier for convenience. Initialize $w^0$ such that $q^0$ and $p^0$ are uniform. Then Exp3 proceeds for every $t \in [T]$ as follows:

1. Draw class $y^t \sim p^t$
2. Observe a scalar reward $L_{y^t}^t$
3. Construct estimator $\widetilde{L}_y^t = L_{y^t}^t \mathbb{1}_{\{y=y^t\}} / p_y^t \ \forall y$
4. Update distribution
   $w^{t+1} = w^t - \eta \widetilde{L}^t$
   $q_y^{t+1} = \exp(w_y^{t+1}) / \sum_{y=1}^k \exp(w_y^{t+1}) \ \forall y$
   $p_y^{t+1} = \gamma \frac{1}{k} + (1-\gamma)q_y^{t+1} \ \forall y$

We can bound the regret of Exp3 (Auer et al., 2002) in our setting, even when the mixing parameter $\gamma$ is not small as otherwise usually required, by following a similar argument as Shalev-Shwartz & Wexler (2016).

For this we will use the relationship between $p$ and $q$ throughout. From $p_y = \frac{\gamma}{k} + (1-\gamma)q_y$ it can easily be verified that,

$$\frac{1}{p_y} \leq \frac{k}{\gamma} \quad \text{and} \quad \frac{q_y}{p_y} \leq \frac{1}{1-\gamma} \quad \text{for all } y. \tag{11}$$

**Lemma 1.** *If Exp3 is run on bounded rewards $L^t_{y^t} \in [0,1]$ $\forall t$ with $\eta \le \gamma/k$ then*

$$\mathcal{R}^{\mathrm{adv}}_T(u) = \sum_{t=1}^T \langle u, \widetilde{L}^t \rangle - \sum_{t=1}^T \langle q^t, \widetilde{L}^t \rangle \le \frac{\log k}{\eta} + \frac{\eta k}{(1-\gamma)\gamma} \sum_{t=1}^T L^t_y \tag{12}$$

*Proof.* We can write the estimator vector as $\widetilde{L}^t = \frac{L^t_{y^t}}{p^t_{y^t}} e_{y^t}$ where $e_{y^t}$ is a canonical basis vector. By assumption we have $L^t_{y^t} \in [0,1]$. Combining this with the bound on $1/p_y$ in Equation 11 we have $\widetilde{L}^t_y \le k/\gamma$ for all $y$. This lets us invoke (Shalev-Shwartz et al., 2011, Thm 2.22) as long as the step-size $\eta$ is small enough. Specifically, we have that if $\eta \le 1/\widetilde{L}^t_y \le \gamma/k$

$$\sum_{t=1}^T \langle u, \widetilde{L}^t \rangle - \sum_{t=1}^T \langle q^t, \widetilde{L}^t \rangle \le \frac{\log k}{\eta} + \eta \sum_{t=1}^T \sum_{y=1}^k q^t_y (\widetilde{L}^t_y)^2. \tag{13}$$

By expanding $\widetilde{L}^t_y$ and applying the inequalities from Equation 11 we can get rid of the dependency on $q$ and $p$,

$$\frac{\log k}{\eta} + \eta \sum_{t=1}^T \sum_{y=1}^k q^t_y (\widetilde{L}^t_y)^2 \le \frac{\log k}{\eta} + \eta \sum_{t=1}^T \frac{q^t_{y^t}}{(p^t_{y^t})^2} (L^t_{y^t})^2$$

$$\le \frac{\log k}{\eta} + \eta \sum_{t=1}^T \frac{k}{(1-\gamma)\gamma} (L^t_{y^t})^2 \tag{14}$$

$$\le \frac{\log k}{\eta} + \frac{\eta k}{(1-\gamma)\gamma} \sum_{t=1}^T L^t_{y^t},$$

where the last line uses the boundedness assumption $L^t_y \in [0,1]$ $\forall y$. $\qquad\square$

It is apparent that we need to control the sum of losses. We will do so by assuming that the model admits a mistake bound as in Shalev-Shwartz & Wexler (2016); Shalev-Shwartz et al. (2011).

**Assumption 1.** *For any sequence of classes $(y^1, ..., y^T) \in [k]^T$ and class conditioned sample indices $(i^1, ..., i^T)$ with $i^t \in \mathcal{N}_{y^t}$ the model enjoys the following bound for some $C' < \infty$ and $C = \max\{k \log k, C'\}$,*

$$\sum_{t=1}^T L_{y^t, i^t}(\theta^t) \le C. \tag{15}$$

Recall that we need to bound $L^t_y := L_{y, i^t}(\theta^t)$, where $y$ are picked adversarially and $i^t$ are sampled uniformly, so the above is sufficient.

**Lemma 2.** *If the model satisfies Assumption 1 and algorithm 1 is run on bounded rewards $L_{y^t, i^t}(\theta^t) \in [0,1]$ $\forall t$ with step-size $\eta = \sqrt{\log k/(4kC)}$ and mixing coefficient $\gamma = 1/2$ then*

$$\sum_{t=1}^T \langle u, \widetilde{L}^t \rangle \le 6C. \tag{16}$$

*Proof.* By expanding and rearranging Lemma 1,

$$\sum_{t=1}^T \langle u, \widetilde{L}^t \rangle \le \frac{\log k}{\eta} + \left( \frac{\eta k}{(1-\gamma)\gamma} + \frac{q^t_{y^t}}{p^t_{y^t}} \right) \sum_{t=1}^T L^t_y, \tag{17}$$

as long as $\eta \le \gamma/k$. Then by using our favorite inequality in Equation 11 and under Assumption 1,

$$\frac{\log k}{\eta} + \left( \frac{\eta k}{(1-\gamma)\gamma} + \frac{q^t_{y^t}}{p^t_{y^t}} \right) \sum_{t=1}^T L^t_y \le \frac{\log k}{\eta} + \left( \frac{\eta k}{(1-\gamma)\gamma} + \frac{1}{1-\gamma} \right) C, \tag{18}$$

To maximize $(1-\gamma)\gamma$ we now pick $\gamma = 1/2$ so that,

$$\frac{\log k}{\eta} + \left(\frac{\eta k}{(1-\gamma)\gamma} + \frac{1}{1-\gamma}\right)C \le \frac{\log k}{\eta} + (4\eta k + 2)\,C. \tag{19}$$

For $\eta$, notice that it appears in two of the terms. To minimize the bound respect to $\eta$ we pick $\eta = \sqrt{\log k/(4kC)}$ such that $\frac{\log k}{\eta} = 4\eta kC$, which leaves us with,

$$\frac{\log k}{\eta} + (4\eta k + 2)C \le 2\sqrt{4kC\log k} + 2C. \tag{20}$$

From the first term it is clear that since $C \ge k\log k$ by assumption then the original step-size requirement of $\eta \le 1/2k$ is satisfies. In this case we can additionally simplify further,

$$2\sqrt{4kC\log k} + 2C \le 6C. \tag{21}$$

$\square$

This still only gives us a bound on a stochastic object. We will now relate it to the empirical class conditional risk $\widehat{L}_y(\theta)$ by using standard concentration bounds. To be more precise, we want to show that by picking $u = e_y$ in $\langle u, \widetilde{L}^t\rangle$ we concentrate to $\widehat{L}_y(\theta)$. Following Shalev-Shwartz & Wexler (2016) we adopt their use of a Bernstein's type inequality.

**Lemma 3** (e.g. Audibert et al. (2010, Thm. 1.2)). *Let $A_1,...,A_T$ be a martingale difference sequence with respect to a Markovian sequence $B_1,...,B_T$ and assume $|A_t| \le V$ and $\mathbb{E}\left[A_t^2 \mid B_1,\ldots,B_t\right] \le s$. Then for any $\delta \in (0,1)$,*

$$\mathbb{P}\left(\frac{1}{T}\sum_{t=1}^{T} A_t \le \frac{\sqrt{2s\log(1/\delta)}}{\sqrt{T}} + \frac{V\log(1/\delta)}{3T}\right) \ge 1 - \delta. \tag{22}$$

**Lemma 4.** *If algorithm 1 is run on bounded rewards $L_{y^t,i^t}(\theta^t) \in [0,1]$ $\forall t$ with step-size $\eta = \sqrt{\log k/(4kC)}$, mixing coefficient $\gamma = 1/2$ and the model satisfies Assumption 1, then for any $y \in [k]$ we obtain have the following bound with probability at least $1 - \delta/k$,*

$$\frac{1}{T}\sum_{t=1}^{T}\widehat{L}_y\left(\theta^t\right) \le \frac{6C}{T} + \frac{\sqrt{4k\log(k/\delta)}}{\sqrt{T}} + \frac{(1+2k)\log(k/\delta)}{3T}. \tag{23}$$

*Proof.* Pick any $y \in [k]$ and let $u = e_y$ in $\langle u, \widetilde{L}^t\rangle$. By construction the following defines a martingale difference sequence,

$$A_t = \widehat{L}_y(\theta^t) - \langle e_y, \widetilde{L}^t\rangle = \widehat{L}_y(\theta^t) - \langle e_y, \frac{1}{p_{y^t}^t}L_{y^t,i^t}(\theta^t)e_{y^t}\rangle. \tag{24}$$

In particular note that $i^t$ is uniformly sampled. To apply the Bernstein's type inequality we just need to bound $|A_t|$ and $\mathbb{E}\left[A_t^2 \mid q^t, \theta^t\right]$. For $|A_t|$ we can crudely bound it as,

$$|A_t| \le |\widehat{L}_y(\theta^t)| + |\langle e_y, \frac{1}{p_{y^t}^t}L_{y^t,i^t}(\theta^t)e_{y^t}\rangle| \le 1 + \frac{k}{\gamma}. \tag{25}$$

To bound the variance observe that we have the following:

$$\mathbb{E}\left[\langle e_y, \widetilde{L}^t\rangle^2 \mid q^t, \theta^t\right] \le \sum_{y'=1}^{k}\frac{p_{y'}^t}{(p_{y'}^t)^2}\mathbb{E}\left[L_{y',i}(\theta^t)^2 \mid \theta^t\right](e_y)_{y'} \quad \text{(with } i \sim \text{unif}(|\mathcal{N}_{y'}|)\ \forall y')$$

$$= \frac{1}{p_y^t}\mathbb{E}\left[L_{y,i}\left(\theta^t\right)^2 \mid \theta^t\right]$$

$$\le \frac{1}{p_y^t} \le \frac{k}{\gamma}. \quad \text{(by Equation 11)}$$

It follows that $\mathbb{E}\left[A_t^2 \mid q^t, \theta^t\right] \le k/\gamma$

Invoking Lemma 3 we get the following bound with probability at least $1 - \delta/k$,

$$\frac{1}{T}\sum_{t=1}^{T}\widehat{L}_y\left(\theta^t\right) \le \frac{1}{T}\sum_{t=1}^{T}\langle e_y, \widetilde{L}^t\rangle + \frac{\sqrt{2\frac{k}{\gamma}\log\left(k/\delta\right)}}{\sqrt{T}} + \frac{(1+\frac{k}{\gamma})\log\left(k/\delta\right)}{3T} \qquad (26)$$

By bounding the first term on the right hand side with Lemma 2 and taking $\gamma = 1/2$ we obtain,

$$\frac{1}{T}\sum_{t=1}^{T}\widehat{L}_y\left(\theta^t\right) \le \frac{6C}{T} + \frac{\sqrt{4k\log\left(k/\delta\right)}}{\sqrt{T}} + \frac{(1+2k)\log\left(k/\delta\right)}{3T} \qquad (27)$$

This completes the proof. $\qquad\square$

We are now ready to state the main theorem.

**Theorem 1.** *If algorithm 1 is run on bounded rewards $L_{y^t,i^t}(\theta^t) \in [0,1]$ $\forall t$ with step-size $\eta = \sqrt{\log k/(4kC)}$, mixing parameter $\gamma = 1/2$ and the model satisfies Assumption 1, then after $T$ iterations with probability at least $1 - \delta$,*

$$\max_{y\in[k]}\frac{1}{n}\sum_{j=1}^{n}\widehat{L}_y\left(\theta^{t_j}\right) \le \frac{6C}{T} + \frac{\sqrt{4k\log(2k/\delta)}}{\sqrt{T}} + \frac{(1+2k)\log(2k/\delta)}{3T} + \frac{\sqrt{2\log(2k/\delta)}}{\sqrt{n}} + \frac{2\log(2k/\delta)}{3n}, \qquad (28)$$

*for an ensemble of size $n$ where $t_j \overset{iid}{\sim} \mathrm{unif}(T)$ for $j \in [n]$.*

*Proof.* If we fix $y$ and let $t_j \overset{iid}{\sim} \mathrm{unif}(T)$ , then the following is a martingale difference sequence,

$$A_j = \widehat{L}_y\left(\theta^{t_j}\right) - \frac{1}{T}\sum_{t=1}^{T}\widehat{L}_y\left(\theta^t\right), \qquad (29)$$

for which it is easy to see that $|A_j| \le 2$ and $A_j^2 \le 1$ given boundedness of the loss. This readily let us apply Lemma 3 with high probability $1 - \delta/2k$. Combining this with the bound of Lemma 4 with probability $1 - \delta/2k$ by using a union bound completes the proof. $\qquad\square$

## B CVAR

Conditional value at risk (CVaR) is the expected loss conditioned on being larger than the $(1-\alpha)$-quantile. This has a distributional robust interpretation which for a discrete distribution can be written as,

$$\mathrm{CVaR}_\alpha\left(\theta, P_0\right) := \sup_{p\in\Delta_m}\left\{\sum_{i=1}^{m}p_i\ell\left(\theta, x_i\right) \text{ s.t. } \|p\|_\infty \le \frac{1}{\alpha m}\right\}. \qquad (30)$$

The optimal $p$ of the above problem places uniform mass on the tail. Practically we can compute this best response by sorting the losses $\{\ell(\theta, x_i)\}_i$ in descending order and assigning $\frac{1}{\alpha m}$ mass per index until saturation.

**Primal CVaR** When $m$ is large a stochastic variant is necessary. To obtain a stochastic subgradient for the model parameter, the naive approach is to compute a stochastic best response over a mini-batch. This has been studied in detail in (Levy et al., 2020).

**Dual CVaR** An alternative formulation relies on strong duality of CVaR originally showed in (Rockafellar et al., 2000),

$$\mathrm{CVaR}_\alpha\left(\theta, P_0\right) = \inf_{\lambda\in\mathbb{R}}\left\{\lambda + \frac{1}{\alpha m}\sum_{i=1}^{m}\left(\ell\left(\theta, x_i\right) - \lambda\right)_+\right\}. \qquad (31)$$

In practice, one approach is to jointly minimize $\lambda$ and the model parameters by computing stochastic gradients as done in (Curi et al., 2019). Alternatively, we can find a close form solution for $\lambda$ under a mini-batch (see e.g. (Xu et al., 2020, Appendix I.1)).

In comparison CFOL acts directly on the probability distribution similarly to primal CVaR. However, instead of finding a best response on a uniformly sampled mini-batch it updates the weights iteratively and samples accordingly (see algorithm 1). Despite this difference, it is interesting that a direct connection can be established between the uncertainty sets of the methods. The following section is dedicated to this.

To be pedantic it is worth pointing out a minor discrepancy when applying CVaR in the context of deep learning. Common architectures such as ResNets incorporates batch normalization which updates the running mean and variance on the forward pass. The implication is that the entire mini-batch is used to update these statistics despite the gradient computation only relying on the worst subset. This makes implementation in this setting slightly more convoluted.

### B.1 Relationship with CVaR

Exp3 is run with a uniform mixing to enforce exploration. This turns out to imply the necessary and sufficient condition for CVaR. We make this precise in the following lemma:

**Lemma 5.** *Let $p$ be a uniform mixing $p = \gamma \frac{1}{m} + (1 - \gamma)q$ with an arbitrary distribution $q \in \Delta_m$ and $\epsilon \in [0, 1]$ such that the distribution has a lower bound on each element $p_i \geq \gamma \frac{1}{m}$. Then an upper bound is implicitly implied on each element, $\|p\|_\infty := \max_{i=[m]} p_i \leq 1 - \frac{(m-1)\gamma}{m}$.*

*Proof.* Consider $p_i$ for any $i$. At least $(m-1)\frac{\gamma}{m}$ of the mass must be on other components so $p_i \leq 1 - \frac{(m-1)\gamma}{m}$. The result follows. $\square$

**Corollary 1.** *Consider the uncertainty set of Exp3,*

$$\mathcal{U}^{\text{Exp3}}(P_0) = \left\{ p \in \Delta_m \mid p_i \geq \gamma \frac{1}{m} \ \forall i \right\}. \tag{32}$$

*Given that the above lower bound is only introduced for practical reasons, we might as well consider an instantiation of CVaR which turns out to be a proper relaxation,*

$$\mathcal{U}^{\text{CVaR}}(P_0) = \left\{ p \in \Delta_m \mid \|p\|_\infty \leq \frac{1}{\alpha m} \right\} \tag{33}$$

*with $\alpha = \frac{1}{(1-\gamma)m+\gamma}$. This leaves us with the following primal formulation,*

$$\text{CVaR}_\alpha(\theta, P_0) := \sup_{p \in \Delta_m} \left\{ \sum_{i=1}^m p_i \ell(\theta, x_i) \ \text{s.t.} \ \|p\|_\infty \leq \frac{1}{\alpha m} \right\}. \tag{34}$$

*Proof.* From Lemma 5 and since CVaR requires $\|p\|_\infty \leq \frac{1}{\alpha m}$ we have,

$$\|p\|_\infty \leq 1 - \frac{(m-1)\gamma}{m} =: \frac{1}{\alpha m} \tag{35}$$

By simple algebra we have,

$$\alpha = \frac{1}{(1-\gamma)m + \gamma}, \tag{36}$$

which completes the proof. $\square$

So if the starting point of the uncertainty set is the simplex and the uniform mixing in Exp3 is therefore only for tractability reasons, then we might as well minimize the CVaR objective instead. This could even potentially lead to a more robust solution as the uncertainty set is larger (since the upper bound in $\mathcal{U}^{\text{CVaR}}$ does not imply the lower bound in $\mathcal{U}^{\text{Exp3}}$).

There are two things to keep in mind though. First, $\alpha$ should not be too small since the optimization problem gets harder. It is informative to consider the case where $\gamma = 1/2$ such that $\alpha = \frac{2}{m+1}$. From this it becomes clear that the recasting as CVaR only works for small $m$. Secondly, despite the uncertainty sets being related, the training dynamics, and thus the obtained solution, can be drastically different, as we also observe experimentally in Section 5. For instance CVaR is known

Table 4: Summary of methods where $N$ is the number of samples and $k$ is the number of classes. The discrete distribution $p$ either governs the distribution over all $N$ samples or over the $k$ classes depending on the algorithm (see columns).

| Uncertainty set | Over data point ($m = N$) | Over class labels ($m = k$) |
|---|---|---|
| $\mathcal{U}^{\mathrm{Exp3}}(P_0) = \left\{ p \in \Delta_m \mid p_i \geq \varepsilon \frac{1}{m} \; \forall i \right\}$ | FOL (Shalev-Shwartz & Wexler, 2016) | CFOL (ours) |
| $\mathcal{U}^{\mathrm{CVaR}}(P_0) = \left\{ p \in \Delta_m \mid \|p\|_\infty \leq \frac{1}{\alpha m} \right\}$ | CVaR (Levy et al., 2020) | LCVaR (Xu et al., 2020) |

for having high variance (Curi et al., 2019) while the uniform mixing in CFOL prevents this. It is worth noting that despite this uniform mixing we are still able to show convergence for CFOL in terms of the worst class in Section 3.1.

In Table 4 we provide an overview of the different methods induced by the choice of uncertainty set.

## C  EXPERIMENTS

(a) $\|\delta\|_\infty = 1/255$           (b) $\|\delta\|_\infty = 2/255$

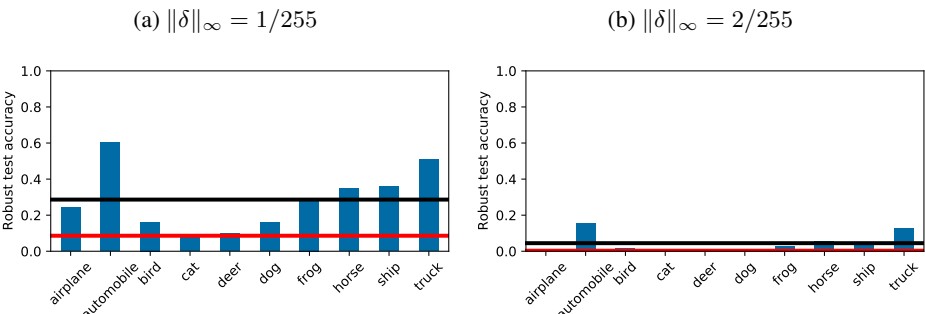

Figure 4: Robust test accuracy under different norm-ball sizes after *clean* training on CIFAR10. The non-uniform distribution over class accuracies, even after clean training, indicates that the inferior performance on some classes is not a consequence of adversarial training. Rather, the inhomogeneity after perturbation seems to be an inherent feature of the dataset.

Table 5: Robust accuracy after early stopping ERM-AT based on different metrics of the validation set. The indicated performance is on the test set. Notice that we only improve the worst class marginally while suffering significantly in terms of average accuracy if we early stop based on the worst class.

|  | Early stopping metrics | |
|---|---|---|
|  | Average robust accuracy | Worst class robust accuracy |
| Average | 0.47 | 0.38 |
| Worst class | 0.14 | 0.18 |

Table 6: Comparison between standard adversarial training (ERM-AT) with and without temporal ensembling (TE). TE improves the robust accuracy on the worst class while maintaining the average robust accuracy. However, TE leads to a drop in terms of the clean accuracy.

|  |  | **ERM-AT** | **ERM-AT (TE)** |
|---|---|---|---|
| $\mathrm{acc}_{\mathrm{clean}}$ | Average | **0.82** | 0.74 |
|  | Worst class | **0.54** | 0.51 |
| $\mathrm{acc}_{\mathrm{rob}}$ | Average | 0.47 | **0.47** |
|  | Worst class | 0.14 | **0.23** |

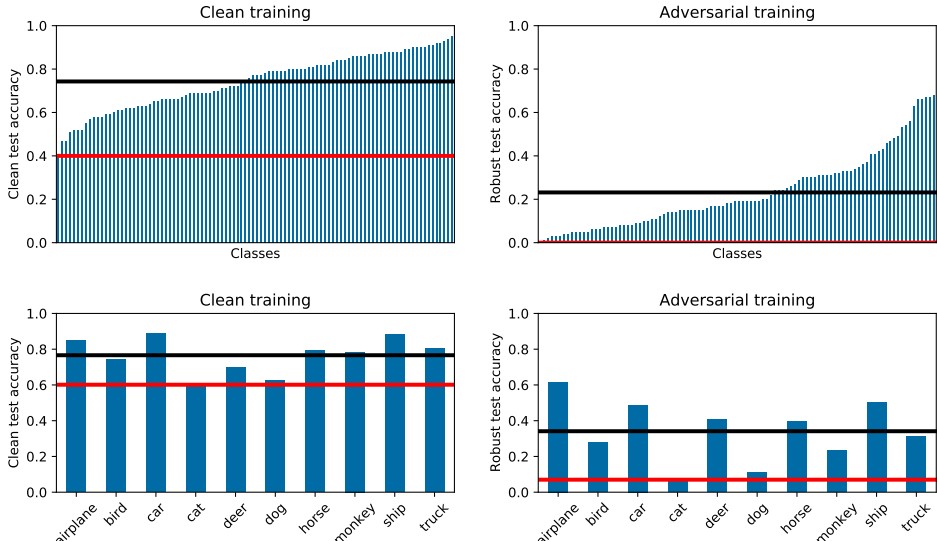

Figure 5: Clean training and adversarial training on CIFAR100 (top) and STL10 (bottom) using ERM-AT with PGD-7 attacks at train time and PGD-20 attacks at test time for the robust test accuracy. The CIFAR100 classes are sorted for convenience. Notice that CIFAR100 has a class with zero robust accuracy with adversarial training.

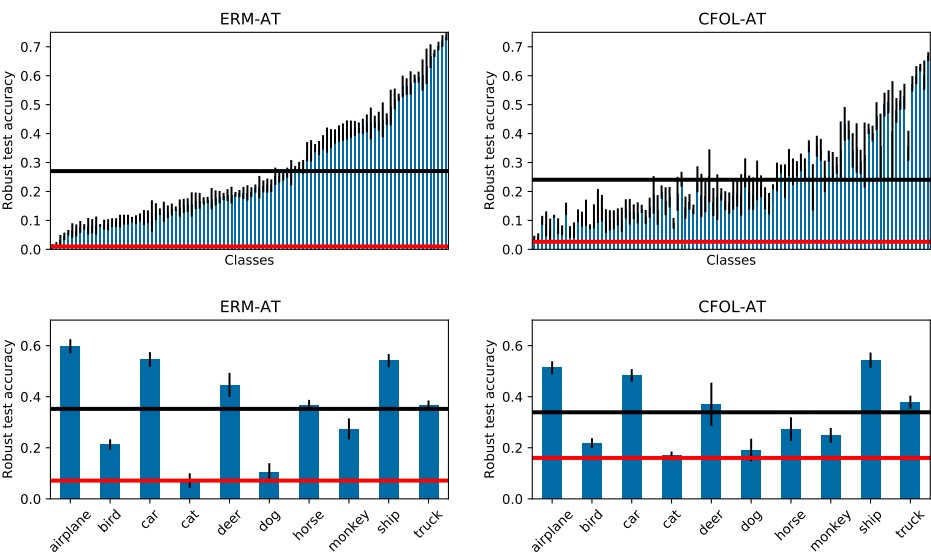

Figure 6: Robust class accuracy for CIFAR100 and STL10 respectively. Vertical black error bars indicate one standard deviation. The red and black horizontal line indicates the minimum and average respectively. The classes on both ERM-AT and CFOL-AT are ordered according to the accuracy on ERM to make comparison easier.

Table 7: For fair comparison we also consider *early stopping based on the worst class accuracy* on the hold-out set. As can be observed the results for CFOL-AT do not differ significantly from early stopping using the average robust accuracy, so standard training setups do not have to be modified further.

|  |  |  | ERM-AT | CFOL-AT | FOL-AT |
|---|---|---|---|---|---|
| CIFAR10 | $\mathrm{acc_{clean}}$ | Average | $0.70 \pm 0.05$ | $0.73 \pm 0.01$ | $\mathbf{0.75} \pm 0.01$ |
|  |  | 20% tail | $0.53 \pm 0.04$ | $\mathbf{0.64} \pm 0.04$ | $0.59 \pm 0.02$ |
|  |  | Worst class | $0.52 \pm 0.03$ | $\mathbf{0.62} \pm 0.03$ | $0.57 \pm 0.01$ |
|  | $\mathrm{acc_{rob}}$ | Average | $0.44 \pm 0.04$ | $0.46 \pm 0.01$ | $\mathbf{0.48} \pm 0.01$ |
|  |  | 20% tail | $0.25 \pm 0.01$ | $\mathbf{0.33} \pm 0.03$ | $0.29 \pm 0.02$ |
|  |  | Worst class | $0.24 \pm 0.01$ | $\mathbf{0.31} \pm 0.02$ | $0.27 \pm 0.02$ |
| CIFAR100 | $\mathrm{acc_{clean}}$ | Average | $\mathbf{0.55} \pm 0.01$ | $0.51 \pm 0.04$ | $0.55 \pm 0.01$ |
|  |  | 20% tail | $0.33 \pm 0.04$ | $\mathbf{0.34} \pm 0.06$ | $0.33 \pm 0.04$ |
|  |  | Worst class | $0.18 \pm 0.02$ | $\mathbf{0.19} \pm 0.04$ | $0.17 \pm 0.03$ |
|  | $\mathrm{acc_{rob}}$ | Average | $\mathbf{0.27} \pm 0.00$ | $0.24 \pm 0.02$ | $0.26 \pm 0.01$ |
|  |  | 20% tail | $0.07 \pm 0.02$ | $\mathbf{0.09} \pm 0.03$ | $0.07 \pm 0.02$ |
|  |  | Worst class | $0.01 \pm 0.01$ | $\mathbf{0.02} \pm 0.02$ | $0.01 \pm 0.00$ |
| STL10 | $\mathrm{acc_{clean}}$ | Average | $0.50 \pm 0.05$ | $\mathbf{0.54} \pm 0.03$ | $0.54 \pm 0.02$ |
|  |  | 20% tail | $0.28 \pm 0.11$ | $\mathbf{0.42} \pm 0.08$ | $0.35 \pm 0.10$ |
|  |  | Worst class | $0.23 \pm 0.09$ | $\mathbf{0.38} \pm 0.09$ | $0.29 \pm 0.09$ |
|  | $\mathrm{acc_{rob}}$ | Average | $0.31 \pm 0.03$ | $\mathbf{0.33} \pm 0.01$ | $0.33 \pm 0.03$ |
|  |  | 20% tail | $0.10 \pm 0.04$ | $\mathbf{0.20} \pm 0.02$ | $0.14 \pm 0.03$ |
|  |  | Worst class | $0.08 \pm 0.04$ | $\mathbf{0.17} \pm 0.05$ | $0.11 \pm 0.02$ |

Table 8: Comparison between different sizes of $\ell_\infty$-ball attacks on CIFAR10. The same constraint is used at both training and test time. When the attack size is increased beyond the usual $8/255$ constraint we still observe that CFOL-AT increases the robust accuracy for the weakest classes while taking a minor drop in the average robust accuracy. Interestingly, the gap between ERM-AT and CFOL-AT seems to enlarge. See Appendix C.2 for more detail on the attack hyperparameters.

|  |  |  | ERM-AT | CFOL-AT |
|---|---|---|---|---|
| $\|\delta\|_\infty \leq 8/255$ | $\mathrm{acc_{clean}}$ | Average | $0.73 \pm 0.01$ | $\mathbf{0.75} \pm 0.00$ |
|  |  | 20% tail | $0.51 \pm 0.03$ | $\mathbf{0.66} \pm 0.01$ |
|  |  | Worst class | $0.47 \pm 0.03$ | $\mathbf{0.63} \pm 0.02$ |
|  | $\mathrm{acc_{rob}}$ | Average | $\mathbf{0.47} \pm 0.03$ | $0.46 \pm 0.00$ |
|  |  | 20% tail | $0.24 \pm 0.02$ | $\mathbf{0.31} \pm 0.02$ |
|  |  | Worst class | $0.20 \pm 0.02$ | $\mathbf{0.30} \pm 0.01$ |
| $\|\delta\|_\infty \leq 12/255$ | $\mathrm{acc_{clean}}$ | Average | $0.63 \pm 0.01$ | $\mathbf{0.63} \pm 0.01$ |
|  |  | 20% tail | $0.31 \pm 0.02$ | $\mathbf{0.53} \pm 0.01$ |
|  |  | Worst class | $0.26 \pm 0.02$ | $\mathbf{0.51} \pm 0.01$ |
|  | $\mathrm{acc_{rob}}$ | Average | $\mathbf{0.38} \pm 0.00$ | $0.35 \pm 0.00$ |
|  |  | 20% tail | $0.12 \pm 0.01$ | $\mathbf{0.23} \pm 0.02$ |
|  |  | Worst class | $0.09 \pm 0.01$ | $\mathbf{0.22} \pm 0.02$ |

Table 9: Model performance on CIFAR10 under AutoAttack (Croce & Hein, 2020). The models still uses 7 steps of PGD at training time with a $\ell_\infty$-constraint of $8/255$. Only at test time is the attack exchanged with AutoAttack under the same constraint. CFOL-AT is robust to AutoAttack in the sense that the worst class performance is still improved. However, as expected, the performance is worse for both methods in comparison with their respective 20-step PGD based attacks at test time.

|  |  | ERM-AT | CFOL-AT |
|---|---|---|---|
| $\text{acc}_\text{clean}$ | Average | $0.74 \pm 0.01$ | $\mathbf{0.75} \pm 0.00$ |
|  | 20% tail | $0.52 \pm 0.03$ | $\mathbf{0.66} \pm 0.01$ |
|  | Worst class | $0.48 \pm 0.04$ | $\mathbf{0.63} \pm 0.02$ |
| $\text{acc}_\text{rob}$ | Average | $\mathbf{0.41} \pm 0.03$ | $0.40 \pm 0.00$ |
|  | 20% tail | $0.17 \pm 0.02$ | $\mathbf{0.21} \pm 0.02$ |
|  | Worst class | $0.12 \pm 0.01$ | $\mathbf{0.20} \pm 0.01$ |

Table 10: Reweighted variant of CFOL. Algorithm 1 samples from the adversarial distribution $p$. Alternatively one can sample data points uniformly and instead reweight the gradients for the model using $p$. In expectation, an update of these two schemes are equivalent. To see why, observe that in CFOL the model has access to the gradient $\nabla_\theta L_{y,i}(\theta)$. We can obtain an unbiased estimator by instead reweighting a uniformly sampled class, i.e. $\mathbb{E}_{y \sim p,i} [\nabla_\theta L_{y,i}(\theta)] = \mathbb{E}_{y \sim \text{unif}(k),i} [k p_y \nabla_\theta L_{y,i}(\theta)]$. With classes sampled uniformly the unbiased estimator for the adversary becomes $\widetilde{L}_{y'} = \mathbb{1}_{\{y'=y\}} L_y k \; \forall y'$. Thus, one update of CFOL and the reweighted variant are equivalent in expectation. However, note that we additionally depended on the internals of the model's update rule and that the immediate equivalence we get is only in expectation. We test the reweighted variant of CFOL-AT on CIFAR10 and observe similar results as for CFOL-AT.

|  | $\text{acc}_\text{clean}$ | | | $\text{acc}_\text{rob}$ | | |
|---|---|---|---|---|---|---|
|  | Average | 20% tail | Worst class | Average | 20% tail | Worst class |
| CFOL-AT (reweighted) | 0.75 | 0.64 | 0.64 | 0.45 | 0.31 | 0.31 |

## C.1 HYPERPARAMETERS

In this section we provide additional details to the hyperparameters specified in Section 5.

For all experiment we use data augmentation in the form of random cropping, random horizontal flip, color jitter and 2 degrees random rotations. For temporal ensembling (TE) we use the parameters in (Dong et al., 2021) and set the momentum to $0.9$, balancing weight parameter to $30$ along a Gaussian ramp-up curve until epoch $50$. See Appendix C.3 for precise definition of the regularization technique.

For CFOL-AT we used the adversarial step-size $\eta = 5 \cdot 10^{-6}$ across all datasets and the same parameters as for ERM-AT described in Section 5. CFOL-AT seems to be reasonable robust to step-size choice as seen in Table 11. Similarly for FOL-AT we use the same parameters as for ERM-AT and set $\eta = 5 \cdot 10^{-7}$ after optimizing based on the worst class robust accuracy. With both methods we make an exception with STL10, where, due to the fewer iterations induced by the 5 times smaller dataset, we scale the adversarial step-size linearly by the dataset ratio 5, such that an equally strong attack can be obtained.

The adversarial step-size picked for both FOL-AT and CFOL-AT is smaller in practice than theory suggests. This suggests that the mistake bound in Assumption 1 is not satisfied for any sequence. Instead we rely on the sampling process to be only mildly adversarial initially as implicitly enforced by the small adversarial step-size. It is an interesting future direction to incorporate this implicit tempering directly into the mixing parameter $\gamma$ instead.

For LCVaR-AT we first optimized over the size of the uncertainty set by adjusting the parameter $\alpha$. Despite getting reasonable performance for some run with $\alpha = 0.2$ the method turned out to have large variance. For this reason we explored reducing the model learning rate $\tau$ for which we found that $\tau = 0.05$ gave better performance while having much smaller variance. It is not surprising that

the method requires smaller step-sizes given how CVaR methods assign higher weight to a fraction of the batch at every iteration as explained in Appendix B. Maybe unnecessarily meticulous, we also ensure that temporal ensembling indeed improves model performance for LCVaR-AT by also running the method without the regularization. Our instantiation of LCVaR-AT used for comparison thus uses $\alpha = 0.2$ and $\tau = 0.05$.

The hyperparameter exploration can be found in Table 11.

Table 11: Hyperparameter exploration for CFOL-AT, LCVaR-AT and FOL-AT.

| | | Parameters | | | $\mathrm{acc_{rob}}$ | | $\mathrm{acc_{clean}}$ | |
| | TE | $\tau$ | $\eta$ | $\alpha$ | Average | Worst class | Average | Worst class |
|---|---|---|---|---|---|---|---|---|
| CFOL-AT | Yes | 0.1 | $1 \times 10^{-6}$ | - | 0.49 | 0.30 | 0.75 | 0.56 |
| | Yes | 0.1 | $2 \times 10^{-6}$ | - | 0.49 | 0.30 | 0.76 | 0.61 |
| | Yes | 0.1 | $5 \times 10^{-6}$ | - | 0.47 | 0.30 | 0.76 | 0.65 |
| | Yes | 0.1 | $5 \times 10^{-5}$ | - | 0.46 | 0.31 | 0.75 | 0.64 |
| LCVaR-AT | No | 0.1 | - | 0.2 | 0.44 | 0.15 | 0.81 | 0.56 |
| | Yes | 0.01 | - | 0.2 | 0.48 | 0.20 | 0.75 | 0.49 |
| | Yes | 0.05 | - | 0.2 | 0.48 | 0.22 | 0.74 | 0.52 |
| | Yes | 0.1 | - | 0.1 | 0.34 | 0.16 | 0.59 | 0.34 |
| | Yes | 0.1 | - | 0.2 | 0.46 | 0.20 | 0.73 | 0.50 |
| | Yes | 0.1 | - | 0.5 | 0.48 | 0.19 | 0.76 | 0.49 |
| FOL-AT | Yes | 0.1 | $2.5 \times 10^{-7}$ | - | 0.50 | 0.19 | 0.77 | 0.48 |
| | Yes | 0.1 | $5 \times 10^{-7}$ | - | 0.50 | 0.24 | 0.79 | 0.60 |
| | Yes | 0.1 | $1 \times 10^{-6}$ | - | 0.50 | 0.22 | 0.80 | 0.55 |

## C.2 EXPERIMENTAL SETUP

In this section we provide additional details for the experimental setup specified in Section 5. The experiments are conducted on the following three datasets:

**CIFAR10** 50,000 training examples of $32 \times 32$ dimensional images and 10 classes.

**CIFAR100** 50,000 training examples of $32 \times 32$ dimensional images and 100 classes.

**STL10** 5000 training examples of $96 \times 96$ dimensional images and 10 classes.

As noted in Section 5 we use the *average* robust accuracy to early stop the model. In contrast with common practice though, we use a validation set instead of the test set to avoid overfitting to the test set. The class accuracies across the remaining two datasets, CIFAR100 and STL10, can be found in Figure 6. We also include results when the model is early stopped based on the worst class accuracy on the validation set in Table 7. The mean and standard deviation in all tables and figures are computed over 5 runs. However, for CIFAR100 we conduct 6 experiments and pick the top 5 to compute the statistics since all methods had one outlier with significantly worst performance.

A radius of $8/255$ is used for the $\ell_\infty$-constraint attack unless otherwise noted. For training we use 7 steps of PGD and a stepsize of $2/255$. At test time we use 20 steps of PGD with a stepsize of $2.5 \times \frac{8/255}{20}$. For $12/255$-bounded attacks we scale the training stepsize and test stepsize proportionally.

## C.3 TEMPORAL ENSEMBLING

Let $f_\theta : \mathbb{R}^d \to \mathbb{R}^k$ be a model parameterized by $\theta$ that assigns a probability distribution over the $k$ classes for any $x \in \mathbb{R}^d$. Temporal ensembling (Laine & Aila, 2016; Dong et al., 2021) computes a so called ensemble prediction for a given data example $x_i$ by maintaining an average over past predictions, $v_i \leftarrow \tau \cdot v_i + (1 - \tau) \cdot f_\theta(x_i)$, where $\tau$ is the momentum term. Let $\ell(f_\theta(x + \delta), y)$ be the training objective. A regularizing term is added to the training objective that encourages predictions to stay close to $v_i$,

$$\min_\theta \sum_{i=1}^n \max_{\delta \in \mathcal{S}} \left\{ \ell(f_\theta(x_i + \delta), y_i) + \omega(t) \cdot \|f_\theta(x_i + \delta) - v_i\|_2^2 \right\}, \qquad (37)$$

where $\omega(t)$ is the balancing weight parameter. Usually $\omega(t)$ interpolates from $0$ to some constant $\omega$ along a Gaussian rampup curve. That is, $\omega(t) = \omega \cdot \exp\left(-5(1-t)^2\right)$ where $t = \min\{\frac{\text{epoch}}{\text{max epoch}}, 1\}$.

## C.4 IMPLEMENTATION

We provide pytorch pseudo code for how CFOL can be integrated into existing training setups in Listing 1. FOL is similar in structure, but additionally requires associating a unique index with each training example. This allows the sampling method to re-weight each example individually.

For LCVaR we use the implementation of (Xu et al., 2020), which uses the dual CVaR formulation described in Appendix B. More specifically, LCVaR uses the variant which finds a closed form solution for $\lambda$, since this was observed to be both faster and more stable in their work.

```python
from torch.utils.data import DataLoader

sampler = ClassSampler(dataset, gamma=0.5)
dataloader = DataLoader(dataset, ..., sampler=sampler)

...

# Training loop:
for img,y in iter(dataloader):

    # attack img
    # compute gradients and update model
    # compute logits

    adv_loss = logits.argmax(dim=-1) != y
    sampler.batch_update(y, eta * adv_loss)
```

Listing 1: Pseudo code for CFOL.

