# OpenReview forum: "Protect the weak: Class focused online learning for adversarial training"
_ICLR.cc/2022/Conference — ICLR 2022 Submitted_

### Official Review · Reviewer_mvco · 2021-11-03

**Correctness:** 4
**Technical Novelty And Significance:** 3
**Empirical Novelty And Significance:** 3
**Recommendation:** 6
**Confidence:** 4

**Main Review:**

# Strengths
- Although the proposed method builds on existing works (Shalev-Shwartz & Wexler, 2016; Auer et al., 2002), the novelty is relatively high: to my knowledge, there exists no AT method that focuses on the class with the worst accuracy, nor combines online learning for this purpose.
- The paper is well written (best writing in my batch), precise, and I enjoyed reading it. I like that the authors (where convenient) first give intuition and then present the result, which helps to follow more easily.
- It does a nice balance between theoretical analysis and empirical evaluation.
----
# Weaknesses
- The paper does a very nice trade-off between theory and empirical analysis. However, as a potential direction for improvement, the empirical evaluation could be more on par with AT literature, for example:
   - analysis with different radius of the ball, i.e. plots with varying $\epsilon$
   - how robust is the method to other attacks (e.g. AutoAttack), is the catastrophic overfitting improved, etc.
- The main result (Thm 1) is for an ensemble of size n, while (if I understood well) the experiments are with one model. The authors mention that a single model suffices, but if I am correct there are no results with an ensemble. Verifying Thm 1 empirically, e.g. showing how $n$ impacts final performances would connect better the result with the experiments.
----
# Recommendations
- Since Tian et al. (2021) is a very recent work I do not take this into account for my rating, but other than that it might be helpful for the audience to include a comparison with it (either theoretical insight about advantages/disadvantages, or empirical comparison on CIFAR-10).
- The abstract leaves it somewhat unclear what is the online learning aspect of the method (included in the title), maybe adding a sentence would be nice.
- Write the requirement $C \geq k \log k $ in Thm 1 more explicitly as a separate assumption before stating Thm 1.
- Typo: Fig.3 noteable $\rightarrow$ notable
- Since the main empirical results use temporal ensembling (TE) add a detailed description of it in the App. for completeness

----
# Questions
1. Fig. 1: Does the robust test accuracy of clean training (left) follow a non-uniform trend across the classes? It would be interesting to see if this observation from Fig.1 is (not) a result of AT.
2. If I am correct, temporal ensembling (TE) is used for all methods. This raises the question if the proposed method works well only in this setup with TE, and for completeness, it would be nice to point out this to the readers. Could the authors provide results without TE on CIFAR-10?


**Summary Of The Paper:**

This paper points out that there could be applications where the worst class accuracy could be critical -- for example, the class with the worst class accuracy can be the main target for the attacks and that the difference between the worst robust test accuracy of a class and the average robust test accuracy can be large (also pointed out independently in Tian et al. (2021)).

Motivated by this, the paper proposes a solution to use the Exponential-weight algorithm (Auer et al., 2002), where, as the empirical distribution is taken the adversarial distribution over classes, resulting in a method called Class Focused Online Learning (CFOL).
The proposed CFOL method is based on the Focused Online Learning (FOL) (Shalev-Shwartz & Wexler, 2016) method, which does not take the class label into account.

The authors show empirically that CFOL consistently improves the worst class accuracy on CIFAR-10, CIFAR-100, and STL10.
Moreover, the authors provide high probability convergence guarantees for the worst class accuracy.

**Summary Of The Review:**

The paper has several advantages
(i) points out an overlooked problem in AT (to my knowledge),
(ii) proposes a novel method for the problem with guarantees
(iii) does a nice trade-off between theoretical analyses and empirical evaluation, and
(iv) it is well written.
There are two main directions for improvement theoretically and empirically, e.g. better connection between the theoretical result and the empirical analyses, more exhaustive empirical evaluation on par with AT literature. Nonetheless, I found the paper very interesting and I think it is relevant for the ICLR audience.
I am happy to raise my score if I misunderstood and the above comments are arguable, or if the authors provide improvements (see questions and comments above).

---

> ### Author Response · Authors · 2021-11-19
> **Reply to reviewer**
>
>
>
> We thank the reviewer for their comments and first address the two major concerns below.
>
> **More exhaustive empirical evaluation** As requested we have tested the methods in different settings:
>
>
>
> * _Varying attack radius:_ We include additional experiments for the larger attack radius of 12/255 containing 5 executions of both ERM and CFOL.
> * _Robustness to other attacks:_ We evaluate 5 independent executions of ERM and CFOL on AutoAttack.
>
> Under larger attack radius of 12/255 during both training time and test time we observe the following:
>
>
> |     |          	| ERM      	| CFOL      	|
> |-|-|-|-|
> |   | Average  	| $0.63 \pm 0.01$         	| $\mathbf{0 .6 3} \pm 0.01$ |
> |   	$\mathrm{acc}_{\mathrm{clean}}$                                       	| 20% tail | $0.31 \pm 0.02$         	| $\mathbf{0 .5 3} \pm 0.01$ |
> |                            	| Worst class  | $0.26 \pm 0.02$         	| $\mathbf{0 .5 1} \pm 0.01$ |
> | 	| Average  	| $\mathbf{0 .3 8} \pm 0.00$ | $0.35 \pm 0.00$         	|
> |   $\mathrm{acc}_{\mathrm{clean}}$                                           	| 20% tail | $0.12 \pm 0.01$         	| $\mathbf{0 .2 3} \pm 0.02$ |
> |                                              	| Worst class  | $0.09 \pm 0.01$         	| $\mathbf{0 .2 2} \pm 0.02$ |
>
>
> All accuracies drop as expected. Specifically, the worst class robust accuracy of ERM drops from previously 20% to 9%. On the other hand CFOL has an accuracy of 22% in the larger attack radius.
>
> Concerning robustness under AutoAttack at test-time we observe the following:
>
> |                    	|          	| ERM                  	| CFOL                 	|
> |----|-----|----|-------|
> |                    	| Average  	| $0.74 \pm 0.01$         	| $\mathbf{0 .7 5} \pm 0.00$ |
> | $\mathrm{acc}_{\mathrm{clean}}$ | 20% tail | $0.52 \pm 0.03$         	| $\mathbf{0 .6 6} \pm 0.01$ |
> |                    	| Worst class  | $0.48 \pm 0.04$         	| $\mathbf{0 .6 3} \pm 0.02$ |
> |                    	| Average  	| $\mathbf{0 .4 1} \pm 0.03$ | $0.40 \pm 0.00$         	|
> | $\mathrm{acc}_{\mathrm{rob}}$            	| 20% tail | $0.17 \pm 0.02$         	| $\mathbf{0 .2 1} \pm 0.02$ |
> |                    	| Worst class  | $0.12 \pm 0.01$         	| $\mathbf{0 .2 0} \pm 0.01$ |
>
> Similarly, all accuracies drop, while the comparison between ERM and CFOL remains the same. That is, CFOL still consistently improves the weakest classes over ERM. Table 8 and Table 9 contain the results of the experiments and we have commented on them in the experimental section (Section 5).
>
> Concerning the two additional questions:
>
>
>
> * **Does the robust test accuracy of clean training follow a non-uniform trend across the classes?** This is indeed an interesting experiment and we thank the reviewer for the suggestion. We carried out two additional experiments testing the robust accuracy after clean training. It turns out that the robust accuracies are non-uniform as the reviewer suspected. This suggests that the problem is inherent to the dataset rather than being a consequence of adversarial training. The experiments can be found in Figure 4 in the appendix (or for convenience [here](https://i.imgur.com/uD241Z4.jpg)) and a remark has been added to the introduction highlighting the observation. \
>  \
> It should be noted that the starting observation in adversarial examples is that with AT (or other robust methods) the average robust accuracy is zero. This implies that the accuracy for any class must also be zero. So the adversarial attack after clean training has to be even smaller than usual (we concretely tested an $\ell_\infty$-ball of radius 1/255 and 2/255).
>
> * **Does CFOL work without TE?** The section on robust overfitting (Section 4) provides experiments without TE on CIFAR10. Specifically the results show that CFOL still enjoys an increase in robust accuracy for the worst class, but there is a larger reduction on the average robust accuracy.
>
> We incorporate the recommendations:
>
>
>
> * **Added explanation of the online learning perspective in the abstract**
> * **Made C > k log k an assumption** We have absorbed this into the mistake bound assumption for a slightly more general claim. If $C \leq k \log k$ we can still have convergence, however the error now depends on $k \log k$ instead of $C$ (where $C$ here refers to the old $C$ which has been replaced by $C’$ in the updated manuscript).
> * **Added detailed description of temporal ensembling in Appendix C.3.**
> * **Connect the theoretical analysis and empirical observation** We note that the theoretical results are in terms of the empirical risk, while the experiments are concerned with the true risk. While the sum decomposition of $k$ and $C$ in the convergence rate of the empirical risk is still informative, the difference explains why the error in the experiments cannot be driven to zero. We have added a comment before the theorem clarifying this and a comment in the conclusion on the interesting direction of obtaining generalization bounds.

---

> > ### Author Response · Authors · 2021-11-26
> > **2nd response to reviewer mvco**
> >
> > We have conducted a major revision that addresses the concerns raised by the reviewer, so we would kindly request the reviewer to re-assess our evaluation and ask any follow-up questions (if any).

---

> > ### Comment · Reviewer_mvco · 2021-12-10
> > **Response after rebuttal**
> >
> > Thanks for the authors' thorough response and for accordingly updating the paper!
> >
> > My main remaining concern is that of the low clean test accuracy of the baseline/starting empirical setup, as I agree with Reviewer 85vN: indeed, while there is no need for state-of-art results, the fact that the starting baseline clean accuracy is notably low raises the question if the selected setup is representative or relevant. In particular, the baseline's clean test accuracy on CIFAR-10 is 74%, when using TE. This increases the need of showing more elaborative results without TE. Also, if the proposed method is recommended to be used always with TE, given that TE decreases the performances for the baseline, a more appropriate baseline in the main paper would be standard training (without TE). Moreover, if I understood correctly the model for this setup is ResNet 50, thus in my opinion even the 82% accuracy reported in Tab.6 is relatively low (in my experience an enlarged LeNet already gives a bit above 70% clean test accuracy, ResNet-18 gives higher, and here the authors use ResNet-50).
> >
> > Thus, I decided to keep my original score: while I like the combination with online learning for AT and the pointed out worst class robustness problem, I am not confident with the presented results (thus the relevance).

---

### Official Review · Reviewer_wi3v · 2021-11-03

**Correctness:** 3
**Technical Novelty And Significance:** 3
**Empirical Novelty And Significance:** 3
**Recommendation:** 6
**Confidence:** 4

**Main Review:**

Strengths:
- Easy to understand writing
- Straightforward, well-motivated method
- Experimental results correspond with claims
- Appropriate baselines
- Experimental results outperform baselines

Weaknesses:
Clarity:
- "ERM" generally refers to algorithms that perform empirical risk minimization and adversarial training does not perform empirical risk minimization. Canonically ERM also refers to the standard training regime. The standard adv training regime should therefore not be referred to as ERM (this a problem throughout the text).
- Table 2/3: $\mathcal{A}$ canonically refers to an algorithm, so $\mathcal{A}_{\text{rob}}$ is confusing notation (one would assume it refers to a robust training algorithm); this should be renamed to another symbol and perhaps labels could be added to the table for further clarity.
- In Table 2/3 it would be good to cite the methods/name them as they are not standard acronyms people use.
- The writing can is often overly verbose or disorganized. For example, in the DRO section of the related work, other seemingly unrelated techniques are sandwiched between paragraphs specifically about DRO. Section 4 could also be considerably condensed (and renamed, as the section describes an experimental setup choice not an application of the algorithm).
- What do the bounds in Table 2/3 represent exactly?

Related work/baselines:
- It would be good to compare with relevant DRO literature: in [0] the authors propose an online learning technique for optimizing worst-case subgroup error (one instantiation of this could be subgroups = labels).

Small fixups:
- Missing space after (2)

[0] https://arxiv.org/abs/1911.08731

**Summary Of The Paper:**

In this work, the authors present a method for maximizing minimum class-wise accuracy in the $\ell_p$-adversarial setting. The proposed method uses an online learning algorithm (exponential weights) to choose classes adversarially at train-time to sample examples from (in place of randomly choosing examples from the given data distribution). The authors find their algorithm can effectively increase the class-wise worst-case accuracy on a number of standard datasets (CIFAR-10/100, STL10) in the $\ell_\inf, \epsilon=8/255$ threat model.

**Summary Of The Review:**

The paper is easy to understand, presents a straightforward/well-motivated method, and backs claims with corresponding experimental results. The presented methods also outperform the baselines. However, the paper has relatively minor issues with clarity and incomplete related work.

---

> ### Author Response · Authors · 2021-11-19
> **Reply to reviewer**
>
>
> We thank the reviewer for their comments and address the suggested corrections below.
>
>
>
> * **Renaming ERM** We have renamed all methods to include the prefix “AT“ to indicate that adversarial training is involved. So ERM is now called ERM-AT. We have kept “ERM” in the name since the method is still performing empirical risk minimization but on the loss defined as $\ell_{\mathrm{adv}}(\theta, x,y) := \max_{\delta \in \mathcal S} \ell(\theta, x+\delta,y)$ (albeit nonsmooth).
> * **Explained bounds in Table 2/3** The bounds represent one standard deviation. We have included this information under “metrics” in the experimental section.
> * **Compared with DRO** We compared with the DRO method we expect to be the strongest. Namely labeled CVaR which exploits the label information in Table 2.
>
> In addition we have made the following changes:
>
>
>
> * Replaced the symbol $\mathcal A$ used for accuracies in the tables with $\mathrm{acc}$
> * Included method name in Table 2/3
> * Shortened DRO section
> * Renamed section on robust overfitting to “Experimental setup: overcoming robust overfitting”

---

> > ### Comment · Reviewer_wi3v · 2021-11-24
> > **Response**
> >
> > Thank you for the clarification.

---

> > > ### Author Response · Authors · 2021-11-26
> > > **2nd response to reviewer wi3v**
> > >
> > > We will be happy to engage if there are other questions. Given that we addressed the main critique points of the review, we would like to understand if there are any remaining concerns of the reviewer.

---

### Official Review · Reviewer_85vN · 2021-11-04

**Correctness:** 3
**Technical Novelty And Significance:** 2
**Empirical Novelty And Significance:** 3
**Recommendation:** 3
**Confidence:** 4

**Main Review:**

While the proposed method is intuitively reasonable for the main purpose of “protecting the weak”, I have several concerns regarding the motivation of the paper and the empirical results.

1. In terms of the motivation, I do not see a strong motivation to protect the weakest class, given the fact that the average robust accuracy is less than 50%. That said, from the perspective of an adversary, arbitrary picking any class can suffice its attack goal. Perhaps, if we assume attacking difference class incurs a different benefit or cost for an attacker, there may exist a particular user case for protecting the weakest class.

2. The performance of the baseline methods presented in this paper is not the state-of-the-art in adversarial robustness literature. In particular, the baseline method (ERM) for CIFAR-10 presented in Table 2 only achieves around 0.74 in clean accuracy and 0.47 in robust accuracy, whereas the state-of-the-art clean accuracy and robust accuracy are over 0.85 and 0.55, respectively [1, 2]. Even the original method of TRADES [3] you implemented should achieve a much higher accuracy than what you presented in Table 2.

[1] Uncovering the Limits of Adversarial Training against Norm-Bounded Adversarial Examples, Sven Gowal, Chongli Qin, Jonathan Uesato, Timothy Mann, Pushmeet Kohli

[2] Reliable Evaluation of Adversarial Robustness with an Ensemble of Diverse Parameter-free Attacks, Francesco Croce, Matthias Hein

[3] Theoretically principled trade-off between robustness and accuracy, Hongyang Zhang, Yaodong Yu, Jiantao Jiao, Eric Xing, Laurent El Ghaoui, and Michael Jordan.

3. Compared with Focused Online Learning (FOL), I do not see an obvious improvement of using your proposed CFOL method. Both the average clean and robust accuracies of CFOL are less than those of FOL by 0.04. Could you clarify why there is an average performance drop?

Other comments:
1. Theorem 1 provides a convergence guarantee for the worst class loss. Theoretically, if we have enough empirical samples, the proposed method should achieve a very small loss (correspondingly, a high accuracy) if T is sufficiently large. Can you clarify why the empirical results provided in Section 5 seem not reflect such theoretical claim?


**Summary Of The Paper:**

Observing that existing adversarially-trained classifiers have different class-wise model performance, this paper aims to protect the most vulnerable class against an adversary who can smartly pick which class to attack. Built upon on a classical Bandit algorithm and adversarial training method, it proposes an algorithm called “Class Focused Online Learning”, which adaptively reweights the class-wise loss function over the training epochs. In addition, the paper proves a convergence guarantee for the worst class loss and provides empirical evaluations of their method on benchmark datasets.


**Summary Of The Review:**


The proposed method combines an online learning algorithm and an adversarial training approach, which is intuitive and new. However, the motivation for protecting the weak in an adversarial setting is not well-explained and justified in my perspective. In addition, the baseline results presented in the paper are far below the state-of-the-art and there does not seem to be a clear advantage of the proposed method over existing approach. Therefore, I suggest a reject for this paper given the abovementioned concerns.

---

> ### Author Response · Authors · 2021-11-19
> **Reply to reviewer**
>
>
> We thank the reviewer for the comments. The main criticism of the reviewer seems to be because of two concerns that we address below.
>
> The paper is **focused on the robust accuracy for the worst class**. We believe this is an important problem, and as we experimentally validate, the proposed approach consistently improves the metric of robust accuracy over the worst class.  Probably the reviewer refers to state-of-the-art numbers in the average accuracy that is NOT our focus. We make an effort to report the average accuracy to showcase the performance that is typically reported in the literature, but we argue that this average performance **overlooks** the worst class performance. To make the point clearer, we have introduced a sentence in the introduction stating that the average robust accuracy is not the main focus of this work.
>
> The reviewer raised concerns that the implementation of standard adversarial training is not state-of-the-art (what is referred to as ERM-AT in the recent version). We match the robust average accuracy in the literature of our implemented method. Concretely we implement the method of Madry et al. [1] and get 47% robust average accuracy under PGD 20 attacks on CIFAR10. This is similar or better than the 45.8% [Table 2, 1] and 47.04% [Table 5, 2] reported in the literature which also uses PGD 20 attacks at test-time.
>
> The goal here is to focus on the robust accuracy of the worst class. To this end, we pick a widely-used baseline for adversarial training and demonstrate our point. We have clarified what method we are running and the primary objective in Section 4, and Section 5 under “hyperparameters”, “baselines” and “metrics”.
>
> It is indeed true that the clean accuracy, on the other hand, is not as high as in the literature. It seems that temporal ensembling had a negative impact on this metric. We have added Table 6 to make this clear and a cautionary note at the end of Section 4. \
>  \
> [1] Towards Deep Learning Models Resistant to Adversarial Attacks, Aleksander Madry, Aleksandar Makelov, Ludwig Schmidt, Dimitris Tsipras, and Adrian Vladu, [https://arxiv.org/pdf/1706.06083.pdf](https://arxiv.org/pdf/1706.06083.pdf) \
> [2] Theoretically Principled Trade-off between Robustness and Accuracy, Hongyang Zhang, Yaodong Yu, Jiantao Jiao, Eric Xing, Laurent El Ghaoui, and Michael Jordan, [https://arxiv.org/pdf/1901.08573.pdf](https://arxiv.org/pdf/1901.08573.pdf)
>
> We address the remaining concerns one by one below.
>
>
>
> * **Motivation** We agree that the average robustness is still an unsolved problem. However, this should not prevent us from starting to address the more complicated problem of the exacerbated vulnerability of the worst class that exists. The fact that we can improve the accuracy from the original 14% of ERM with TE to 31% with CFOL using TE shows that this is indeed possible to address. \
>  \
> We do not understand why an average robust accuracy below 50% invalidates the study of the weakest class. This is better than random guessing on a 10 class classification problem like CIFAR10, so the model is indeed learning. The average performance of adversarially trained models are even more impressive considering that a model trained with unperturbed examples has zero accuracy. That these models can still perform poorly on a specific class seems important.
> * **Comparison with FOL** Our focus is on the accuracy of the worst class for which CFOL is consistently better than FOL. Consider for example CIFAR10 where the improvement is from 22% to 30% for the robust accuracy on the worst class. It is indeed true that FOL surprisingly improves the average even over ERM, which we comment on in the experimental section. It is definitely a very interesting question to understand this phenomenon as a future direction.
> * **Why can empirical results not have zero error when Theorem 1 implies zero error for $T\rightarrow\infty$?** The theoretical results are for the _empirical_ _risk_. This is still informative of the true risk, as you need to at least be able to drive down the empirical risk. In particular, we expect the sum decomposition of k and C to remain relevant. However, as the reviewer pointed out, the test error is far from zero in adversarial training. So there is a gap which makes generalization bounds a very interesting future direction. We have included two remarks to highlight this: one right before Theorem 1 in Section 3.1 and one in the conclusion.

---

> > ### Comment · Reviewer_85vN · 2021-11-25
> > **Post-Rebuttal Review**
> >
> > Thanks for the response to my concerns. The authors emphasize many times in the response that they focus on the robust accuracy for the worst class, but both the manuscript and the response still do not address my concern: what is the motivation to only focus on the worst class (what is the application scenario for this). In addition, the clean accuracies presented in Table 2 are much lower than the state-of-the-art, which should be explained the reason behind in my perspective. Therefore, after reading the response, I choose to keep my score unchanged.

---

> > > ### Author Response · Authors · 2021-11-26
> > > **2nd response to reviewer 85vN**
> > >
> > >
> > > **Motivation** There are a number of safety-critical applications, such as autonomous driving, where our contribution is important. In such applications, we do not want the models injected into the decision making process to be susceptible to small perturbations for _any_ of the classes. Imagine the task of **street sign classification** for an autonomous vehicle [1,2,3]. Even if we reach 99% robust accuracy it remains a critical problem if an attacker can _consistently_ fool a particular sign. We mention that autonomous driving is an application that can benefit from our proposed method already in sec. 2.
> > >
> > > Another prominent example is that of **medical imaging**. Image reconstruction can be oblivious to small features of the input and additionally introduce artifacts based on small perturbations [4,5]. Note that these perturbations can be naturally occurring, so it remains a problem even if there is no malicious adversary controlling the input of the system, which is not to be expected in a controlled medical setting. In this setting, adversarial training has been used to increase the performance on spatially small features [4]. This makes it an interesting case for our methods, as downstream classification tasks might similarly be affected in a non-uniform way. Even when assuming that only a single class is compromised, this can be fatal if the misclassified class is for instance a type of tumor.
> > >
> > > Even though we present two applications, there are other important cases in safety-critical applications that might be of interest. We simply aim to provide a direct answer to the request of the reviewer for a concrete application.
> > >
> > > **Clean accuracy** Concerning clean accuracy we would like to point out the ICLR guidelines, which explicitly mentions that “a lack of state-of-the-art results does not by itself constitute grounds for rejection”. In addition, we reiterate that we focus on an observation regarding the robust accuracy that we believe is interesting to the community. Specifically, this work demonstrates that it is harder to learn a robust model of some classes than others using standard methods. This problem is regardless of the behavior of clean accuracy. We do however include results on clean accuracy to be consistent with the literature.
> > >
> > > [1]: DARTS: Deceiving Autonomous Cars with Toxic Signs, C Sitawarin, AN Bhagoji, A Mosenia, M Chiang, P Mittal ([https://arxiv.org/pdf/1802.06430.pdf](https://arxiv.org/pdf/1802.06430.pdf))
> > >
> > > [2]: Adaptive Square Attack: Fooling Autonomous Cars With Adversarial Traffic Signs, Y Li, X Xu, J Xiao, S Li, HT Shen ([https://ieeexplore.ieee.org/document/9165820](https://ieeexplore.ieee.org/document/9165820))
> > >
> > > [3]: A collection of easily deployable adversarial traffic sign stickers, H Lengyel, V Remeli, Z Szalay ([https://www.degruyter.com/document/doi/10.1515/auto-2020-0115/pdf](https://www.degruyter.com/document/doi/10.1515/auto-2020-0115/pdf))
> > >
> > > [4]: Addressing The False Negative Problem of Deep Learning MRI Reconstruction Models by Adversarial Attacks and Robust Training, Cheng, Kaiyang, et al. ([http://proceedings.mlr.press/v121/cheng20a/cheng20a.pdf](http://proceedings.mlr.press/v121/cheng20a/cheng20a.pdf))
> > >
> > > [5]: On instabilities of deep learning in image reconstruction and the potential costs of AI, Vegard Antun, Francesco Renna, Clarice Poon, Ben Adcock, and Anders C. Hansen ([https://www.pnas.org/content/117/48/30088](https://www.pnas.org/content/117/48/30088))

---

### Official Review · Reviewer_MFgG · 2021-11-05

**Correctness:** 4
**Technical Novelty And Significance:** 2
**Empirical Novelty And Significance:** 2
**Recommendation:** 3
**Confidence:** 3

**Main Review:**

-I'm not sure what do you mean by saying that the adversary is "choosing the class". There is the magnitude of noise, and the adversary maximizes the average loss of the learner. It doesn't choose a specific class (although some might be more vulnerable than others)

-The error across classes is not perfectly uniform in the adversarial training process, but this is the case (sometimes) in the standard setting. What makes the adversarial training process different?

-In the suggested method, you can perform exponential weights with full information, I don't see the reason for using bandit feedback (and it is not explained).

-The technical novelty is only marginally novel and quite straightforward.

-What about the following baseline - you have the empirical error, so perform an ERM with that "penalize" more some classes.
[I guess the Hedge should work better]

**Summary Of The Paper:**

The setting is robust learning to perturbations at test time (adversarial examples), where instead of minimizing the average loss, the goal is to minimize the loss of the class with the lowest accuracy.
That is, instead of a min-max problem, there is another maximization component over the classes.

This is motivated by an example that shows a larger variance in losses between classes than in the standard setting.

The authors suggest a method from online learning - exponential weight in the bandit setting (that is - Exp3).

**Summary Of The Review:**

My vote is - reject.
I mentioned my concerns. It is possible that I did not assess the contributions correctly, I am willing to read the other reviews and reconsider my score if they convince me otherwise.

---

> ### Author Response · Authors · 2021-11-19
> **Reply to reviewer**
>
>
> We thank the reviewer for the feedback. Allow us to answer the questions of the reviewer and clarify any concerns.
>
>
> * **The adversary’s power** We are not saying that the adversary has control over the class in standard adversarial training. In the usual setting where we are minimizing the average accuracy, the adversary indeed only chooses the perturbation. We observe that the accuracy on the worst class can be much smaller than the average. So if the adversary at _test time_, in addition to the perturbation, could choose the class on which the model is evaluated, then the model would suddenly have almost no robustness. We have made this clearer in the introduction by stating that choosing the class is an _additional_ power of the adversary.
> * **The reason for heterogeneity** This is indeed a very interesting question. We conducted an additional experiment, which measured robust accuracy with an $\ell_\infty$ attack of radius $1/255$ and $2/255$ after clean training. Interestingly, we find that there is, similarly, increased discrepancies between class accuracies, even on this model trained without adversarial perturbations. This shows that the problem is not a consequence of adversarial training, but rather a more general problem when concerned with adversarial examples. We have added experiments in Figure 4 of the updated manuscript.
> * **Bandit feedback** Bandit feedback is necessary for making the method compatible with the mini-batch mode used for neural network training. A truly full information setting is only possible if we could go over all data points at every iteration of the algorithm. This is not feasible in modern deep learning frameworks where typically only a small portion of the data points fit into the GPU. We have clarified this with a sentence under “method” (Section 3).
> * **Suggested baseline** If we understand correctly, the reviewer is suggesting reweighting the gradients of a given example according to some class-dependent penalty. A principled way of doing so is running CFOL but using the learned distribution over classes for reweighting the gradients on uniformly drawn samples instead of drawing samples according to the distribution. We experimented with this alternative at an early stage and reweighting demonstrated an equivalent performance to sample-based CFOL. Let us illustrate why both intuitively and experimentally. \
>  \
> First of all, in expectation, an update of these two schemes are equivalent. To see why, observe that in CFOL the model has access to the gradient $\nabla_\theta L_{y,i}(\theta)$. For simplicity, let us assume that the data distribution has a uniform distribution over classes. We can obtain an unbiased estimator by instead reweighting a uniformly sampled class, i.e. $\mathbb E_{y \sim p, i}\left[\nabla_\theta L_{y,i}(\theta)\right] = \mathbb E_{y \sim \operatorname{unif}, i}\left[k p_y \nabla_\theta  L_{y,i}(\theta)\right]$. With classes sampled uniformly the unbiased estimator for the adversary becomes $\widetilde L_{y'} = 1_{\{y'=y\}} L_{y} k \ \forall y’$. Thus, one update of CFOL and the reweighted variant are equivalent in expectation. However, note that we additionally depended on the internals of the model’s update rule and that the immediate equivalence we get is only in expectation. \
>  \
> The results for reweighted CFOL can be found below. The robust accuracy of the worst class remains within one standard deviation of CFOL while the average robust accuracy is 1% worse. We have included the experiments using reweighting and the explanation of the connection with the sampling version of CFOL in Table 10 of the updated manuscript.
>
>     |                  	| Average | 20% tail | Worst class |
>     |----------------------|---------|--------------|-------------|
>     | CFOL (reweighted) | 0.45	| 0.31     	| 0.31    	|

---

> > ### Author Response · Authors · 2021-11-26
> > **2nd response to reviewer MFgG**
> >
> > We would be happy to reply to any new questions regarding the online learning or how this is applied in our problem setting.

---

### Author Response · Authors · 2021-11-19
**Major revision**

We are thankful to the reviewers for their feedback. Below, we address all comments and questions raised. In particular, we have made the following extensions:



* We have performed an ablation study with different attack radius (Table 8).
* We have performed experiments with AutoAttack (Table 9).
* We have considered a reweighted CFOL version (Table 10).
* We have conducted experiments for the robust accuracy of clean training (Figure 4).

In addition to the aforementioned extensions, the reviewers have already expressed appreciation for the proposed method which they remark as well-motivated and novel (reviewer wi3v, mvco and 85vN). Reviewers wi3v, mvco  remark the paper is well-written and easy to understand, while reviewer mvco advocates this is “relevant for the ICLR audience”.

---

### Decision · Program_Chairs · 2022-01-20

**Decision:**

Reject

**Comment:**

The paper looks at the worst-class adversarial error for multi-class classification problems. The question is given a certain level of adversarial error on average, is it possible that some classes have adversarial error significantly worse than average? And if so, is this a problem? I agree with the authors that there are applications where such an imbalance could be problematic; other than the examples provided by the authors I can also think of this being important from a point of view of fairness, depending on what exactly the class labels represent. The reviewers have raised the question of low accuracies reported in the empirical results compared to the state of the art on those datasets for adversarial learning. I share these concerns -- especially it's worth understanding whether more accurate models also have such an imbalance, or whether this imbalance is a result of incomplete training or models that are not representationally powerful enough. While I agree with the authors that 'state of the art' results' are not required for ICLR submissions, especially those making conceptual contriubtions, in this case I think further experiments may be needed in addition to addressing the other questions raised in the reviews. The authors acknowledge that they have made significant revisions in response to the reviews, but I think that would require a fresh review cycle.